# When Every Millisecond Counts: Real-Time Anomaly Detection via the Multimodal Asynchronous Hybrid Network

Dong Xiao [* 1 2]   Guangyao Chen [* 1]   Peixi Peng [3]   Yangru Huang [1]   Yifan Zhao [4]   Yongxing Dai [5]
Yonghong Tian [1 2 3 6]

## Abstract

Anomaly detection is essential for the safety and reliability of autonomous driving systems. Current methods often focus on detection accuracy but neglect response time, which is critical in time-sensitive driving scenarios. In this paper, we introduce real-time anomaly detection for autonomous driving, prioritizing both minimal response time and high accuracy. We propose a novel multimodal asynchronous hybrid network that combines event streams from event cameras with image data from RGB cameras. Our network utilizes the high temporal resolution of event cameras through an asynchronous Graph Neural Network and integrates it with spatial features extracted by a CNN from RGB images. This combination effectively captures both the temporal dynamics and spatial details of the driving environment, enabling swift and precise anomaly detection. Extensive experiments on benchmark datasets show that our approach outperforms existing methods in both accuracy and response time, achieving millisecond-level real-time performance. The code is available at https://github.com/PKU-XD/EventAD.

## 1. Introduction

Autonomous driving technology has been at the forefront of research and development in recent years, promising to

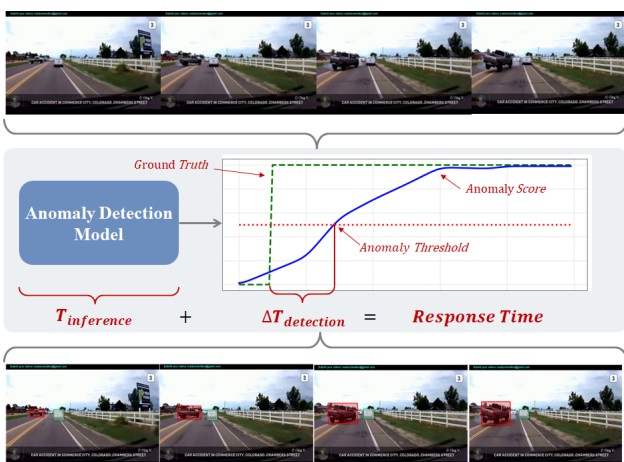

*Figure 1.* For real-time anomaly detection with an emphasis on response time, the overall response time is primarily influenced by the model's inference duration and the time taken to identify anomalies.

revolutionize the transportation industry by enhancing road safety, reducing traffic congestion, and improving fuel efficiency (Huang et al., 2018; Caesar et al., 2020). The core of autonomous vehicles lies in their ability to perceive and interpret complex and dynamic driving environments accurately and efficiently. Among the various perception tasks, anomaly detection plays a pivotal role in ensuring the safety and reliability of autonomous systems. Anomalies, such as unexpected obstacles, erratic behaviors of other road users, or sudden changes in the environment, can pose significant risks if not promptly identified and addressed (Fang et al., 2024). For example, a pedestrian darting onto the road from behind a parked vehicle or a sudden road obstruction requires the autonomous vehicle to react within milliseconds to prevent potential accidents (Gehrig & Scaramuzza, 2024).

Anomaly detection is a vital component in ensuring the safety of autonomous driving systems. Despite substantial progress, current methods often prioritize detection accuracy over an equally crucial factor: *response time* (Cui et al., 2023; Zeng et al., 2023). Many state-of-the-art solutions rely on increasingly sophisticated deep neural networks, which can incur large inference latencies (Yao et al., 2022;

*Equal contribution [1]National Key Laboratory for Multimedia Information Processing, School of Computer Science, Peking University [2]Department of Software and Microelectronics, Peking University [3]School of Electronic and Computer Engineering, Peking University [4]State Key Laboratory of Virtual Reality Technology and Systems, SCSE, Beihang University [5]Baidu Inc [6]Peng Cheng Laboratory. Correspondence to: Guangyao Chen <gy.chen@pku.edu.cn>, Peixi Peng <pxpeng@pku.edu.cn>, Yonghong Tian <yhtian@pku.edu.cn>.

*Proceedings of the 42nd International Conference on Machine Learning*, Vancouver, Canada. PMLR 267, 2025. Copyright 2025 by the author(s).

Karim et al., 2023). In a domain where a delay of even a few hundred milliseconds can dictate the difference between safe braking and a collision (Tian et al., 2024), these lengthy detection times significantly compromise the very safety guarantees that autonomous vehicles are designed to deliver (Wang et al., 2023).

To overcome this limitation, we focus on the task of **real-time anomaly detection** in autonomous driving, specifically targeting the detection of sudden hazardous anomalies from the ego-vehicle perspective, while explicitly incorporating response time into the performance evaluation metrics. In addition to detection accuracy, this perspective emphasizes minimizing the total response time—encompassing both inference latency and the delay between anomaly occurrence and detection (Tian et al., 2024). By centering on this time-critical requirement, our approach seeks to bridge the gap between high-accuracy detection and the urgent demands of real-world driving scenarios, where prompt decision-making is paramount for safety.

To address the challenge of real-time anomaly detection, we introduce a **multimodal asynchronous hybrid network** that reduces inference latency and detection delays while preserving high accuracy. Our approach strategically combines event cameras and conventional RGB cameras to exploit their complementary advantages. Event cameras capture brightness changes at microsecond resolution, providing sparse yet highly informative data for dynamic scenes (Gallego et al., 2020). At the same time, RGB cameras deliver rich spatial context, albeit with higher latency and susceptibility to motion blur (Zhou & Jiang, 2024). By fusing the asynchronous, sparse event streams with continuous image data, we enable the model to capture both fine-grained spatial details and rapid temporal cues.

In particular, we employ an asynchronous Graph Neural Network (GNN) to process the event stream data, harnessing the inherent sparsity and asynchronous nature of event cameras (Li et al., 2021). Meanwhile, a CNN-based ResNet extracts high-level spatial features from the RGB images (He et al., 2016). Subsequently, by leveraging a GRU module to jointly learn the spatio-temporal relationships of object-level and frame-level event streams and RGB features, our model can anticipate anomaly trends in advance, achieving both accurate and rapid anomaly detection. Notably, we are the first to introduce the unique characteristics of event streams as critical features for road traffic anomaly detection, and we fully exploit their high temporal resolution and asynchronous nature through an asynchronous network architecture. This approach is specifically designed for real-time operation, enabling crucial millisecond-level response times in autonomous driving environments. Extensive experiments on multiple benchmark datasets demonstrate that our approach not only outperforms existing methods in de-

tection accuracy but also substantially lowers response time, fulfilling the stringent safety requirements of real-world autonomous driving environments (Karim et al., 2023; Yao et al., 2022).

Our main contributions are summarized as follows:

- We formalize the task of *real-time anomaly detection*, emphasizing the pivotal role of rapid response in safety-critical autonomous driving scenarios. In doing so, we underscore how existing methods overlook this time-sensitive aspect, compromising overall system safety.

- We propose a novel network architecture that synergistically fuses event stream data with RGB images to strike an optimal balance between minimal inference latency and high detection accuracy. By capitalizing on the asynchronous, fine-grained temporal information from event cameras and the rich spatial features of conventional images, our model delivers reliable performance even under challenging conditions.

- We conduct extensive experiments on multiple benchmark datasets, demonstrating that our approach not only surpasses state-of-the-art baselines in detection accuracy but also significantly reduces response time. These findings validate the practical effectiveness of our method, reinforcing its suitability for real-world autonomous driving applications.

## 2. Related Works

**Ego-View Traffic Accident Detection (TAD).** TAD aims to identify accidents within specific time frames and regions using two primary approaches: frame-level and object-level methods. Frame-level methods extract features from video frames and classify them to detect accidents (Vijay et al., 2022; Zhou et al., 2022). For example, You and Han (You & Han, 2020) developed a traffic dataset and utilized a 3D-CNN for accident localization. Reconstruction-based techniques (Chong & Tay, 2017; Zhao et al., 2017; Gong et al., 2019) identify anomalies by comparing reconstructed frames with actual ones. Due to the limited availability of real accident data, synthetic datasets and domain adaptation methods are often employed to enhance performance (Batanina et al., 2019; Tamagusko et al., 2022). Object-level methods focus on the consistency of object movements over time by using detectors and trackers to generate trajectories, thereby reducing the impact of dynamic backgrounds. These methods analyze trajectories (Santhosh et al., 2021; Chakraborty et al., 2018) or evaluate the consistency of object positions (Le et al., 2020; Taccari et al., 2018; Hu et al., 2021a) to detect accidents. Yao *et al.* (Yao et al., 2022; 2019) proposed an unsupervised approach that predicts future object positions, flagging significant deviations

as potential accidents. Additionally, object-level strategies model interactions among objects to detect abrupt contextual changes that may indicate accidents (Fang et al., 2022; Yamamoto et al., 2022; Roy et al., 2020; Vijay et al., 2022). MOVAD (Rossi et al., 2024) achieves efficient online detection of traffic anomalies based solely on dashcam videos by combining a Video Swin Transformer and an LSTM module, and it is the first to introduce the concept of online traffic anomaly detection.

**Ego-View Traffic Accident Anticipation (TAA).** TAA focuses on predicting potential collisions by identifying unusual behaviors in advance, providing critical time for safe decision-making. It primarily involves predicting object trajectories to assess accident likelihood through detection, tracking, and prediction workflows (Haris et al., 2021; Thakur et al., 2024). In complex environments like highways, methods such as SVMs and HMMs analyze vehicle trajectories to forecast accidents (Xiong et al., 2017; Gutierrez-Osorio & Pedraza, 2020). To handle frequent obstructions in dashcam footage, recent approaches employ Dynamic Spatial Attention (DSA) (Chan et al., 2017) and RNNs to capture complex spatial and temporal relationships (Karim et al., 2022). Beyond trajectory prediction, TAA also involves identifying high-risk areas (Karim et al., 2023; Shimomura et al., 2024) and modeling driver attention (Chen et al., 2023). Risk localization techniques integrate agent representations with regional interactions to highlight areas with high accident probabilities (Zeng et al., 2017). The DRAMA dataset (Malla et al., 2023), which combines visual and textual data, enhances prediction accuracy by providing detailed descriptions of potential accident scenarios. Driver attention models utilize gaze direction to emphasize possible dangers, focusing on risky areas (Bao et al., 2021). Additionally, attention maps improve TAA interpretability by highlighting high-risk zones, supporting integrated approaches that combine trajectory prediction, risk localization, and driver attention modeling (Monjurul Karim et al., 2021).

## 3. Real-Time Anomaly Detection

Real-Time Anomaly Detection in autonomous driving is essential for ensuring safety by swiftly identifying and responding to unexpected objects and behaviors in the driving environment. This task demands a system that operates with minimal latency, capable of detecting dynamic changes such as pedestrians suddenly crossing the road or vehicles appearing abruptly. The primary challenge is to achieve high precision in anomaly recognition while maintaining response times at the millisecond level.

**Problem Formulation.** Let $\{X_t\}_{t=1}^{T}$ denote a sequence of sensor observations, where $X_t \in \mathbb{R}^n$ represents the sensor data at time $t$. The objective is to detect anomalies in real-

time, identifying unexpected objects or behaviors that may pose risks.

We define an anomaly indicator function $A_t$ as:

$$A_t = \mathbb{I}(X_t \text{ is anomalous}), \tag{1}$$

where $\mathbb{I}(\cdot)$ is the indicator function, returning 1 if the condition is true and 0 otherwise.

A detection model $f(\cdot)$ assigns an anomaly score $s_t = f(X_t)$. An anomaly is detected when the score exceeds a threshold $\theta$:

$$\hat{A}_t = \mathbb{I}(s_t > \theta). \tag{2}$$

**Response Time.** Response time $R$ is a critical metric, comprising the detection delay and the model's inference time:

$$R = \Delta T_{\text{detection}} + T_{\text{inference}}, \tag{3}$$

where $\Delta T_{\text{detection}} = T_{\text{detection}} - T_{\text{occurrence}}$ is the delay between the anomaly occurrence and its detection. $T_{\text{inference}}$ is the time taken by the model to process the input and produce a result. The goal of Real-Time Anomaly Detection is to minimize $R$ while ensuring high detection accuracy by:

- **Minimizing Inference Time** ($T_{\text{inference}}$): Developing efficient algorithms that process data rapidly to reduce computational delays.

- **Reducing Detection Delay** ($\Delta T_{\text{detection}}$): Enhancing the model's ability to promptly identify anomalies immediately after they occur.

## 4. Method

To achieve real-time anomaly detection with minimal inference time and detection delays, we propose a multimodal asynchronous hybrid network that integrates sparse event streams with RGB image data. Our framework processes RGB images using a ResNet to extract appearance features and captures event data through an asynchronous GNN with spline convolution. The image features are shared unidirectionally with the GNN, enabling the GNN to enhance event feature representation without reciprocal communication. This design significantly improves performance, especially in scenarios with sparse events, such as static or slow-moving conditions.

The features from both modalities are fused and passed to a detection head, which generates object-bounding boxes. These object-level features are further refined using a global graph that incorporates bounding box priors, enhancing spatial anomaly detection. To capture temporal dependencies, we employ a Gated Recurrent Unit (GRU) that processes the video sequences. An attention mechanism assigns higher weights to potentially anomalous objects, ensuring focused

and efficient analysis. The combination of spatial, temporal, and attention-enhanced features enables accurate and swift anomaly detection.

In Sec. 4.1, we describe the network backbone, highlighting the unidirectional integration of image and event features and the role of asynchronous GNNs in feature representation. Sec. 4.2 elaborates on how spatial and temporal features are fused for anomaly detection. Figure 2 presents the overall architecture of our proposed framework.

## 4.1. Multimodal Asynchronous Hybrid Network

To achieve real-time anomaly detection, we propose a Multimodal Asynchronous Hybrid Network that efficiently integrates RGB images and event streams. Our network consists of two parallel branches: a CNN for processing image data and a GNN for handling event streams. This dual-branch architecture enables rapid and accurate feature extraction from both modalities.

**Image Feature Extraction.** The CNN branch, denoted as $F_I$, processes input images $I \in \mathbb{R}^{H \times W \times 3}$ to extract rich spatial features. Utilizing a ResNet architecture, $F_I$ generates detection outputs $D_I$ and intermediate feature maps $G_I = \{g_I^l\}_{l=1}^L$ at various layers. These intermediate features are reused in the GNN branch to enhance computational efficiency.

**Asynchronous Event Graph Construction.** Event streams $E = \{e_i = (x_i, t_i, p_i)\}$, where $x_i = (u_i, y_i)$ denotes pixel coordinates, $t_i$ is the timestamp, and $p_i \in \{-1, 1\}$ represents polarity, are captured by event cameras when luminance changes exceed a threshold $C$:

$$|\Delta L| > C. \qquad (4)$$

These events are modeled as nodes in a graph $G = (V, E)$ with normalized spatial coordinates $\hat{x}_i = \left(\frac{u_i}{W}, \frac{y_i}{H}\right)$ and scaled timestamps $\hat{t}_i = \beta t_i$. Edges are formed based on spatial and temporal proximity within a radius $R$, and edge features are defined as:

$$e_{ij} = \frac{1}{2}(n_{j,xy} - n_{i,xy}) + \frac{1}{2}, \qquad (5)$$

where $n_{i,xy}$ and $n_{j,xy}$ are the normalized spatial coordinates of nodes $i$ and $j$. Each node connects to up to 16 neighbors to maintain computational efficiency.

**Event Feature Extraction.** We employ a Deep Asynchronous Graph Neural Network (DAGr) (Gehrig & Scaramuzza, 2024) to process the event graph using residual graph convolutional layers with spline convolutions:

$$f_i' = W_c f_i + \sum_{j \in \mathcal{N}(i)} W(e_{ij}) f_j, \qquad (6)$$

where $W_c$ and $W(e_{ij})$ are learnable weights, and $\mathcal{N}(i)$ denotes the neighbors of node $i$. Spline convolutions enable efficient aggregation of neighbor information, accelerating computation through lookup tables during deployment. Temporal consistency is maintained by aggregating nodes into a voxel grid and applying directional voxel pooling, which preserves the temporal order of events.

**Feature Fusion.** To integrate the extracted features from both modalities, we fuse the CNN and GNN outputs by augmenting each GNN node feature $f_i$ with the corresponding CNN feature $g_I(\hat{x}_i)$ sampled at the node's location:

$$f_i' = [f_i, g_I(\hat{x}_i)]. \qquad (7)$$

This fusion enhances the model's ability to leverage spatial information from images alongside the temporal dynamics captured by event streams, improving the detection of anomalies in diverse driving scenarios.

The Multimodal Asynchronous Hybrid Network is optimized for real-time performance by utilizing asynchronous processing and efficient feature fusion. This design ensures minimal latency and high accuracy in detecting anomalies, making it well-suited for the stringent requirements of autonomous driving systems.

## 4.2. Anomaly Detection Network

To enable real-time anomaly detection, our Anomaly Detection Network efficiently extracts and processes object-level features from both event streams and RGB images, capturing spatial and temporal dynamics with minimal latency.

**Object Feature Extraction.** We utilize an asynchronous GNN to extract features from event data overlapping with detected bounding boxes. For each object $i$ at time $t$, the GNN generates an event-based feature $o_{t,i}$:

$$o_{t,i} = \text{AsyncGNN}(E_{t,i}; \theta_{\text{GNN}}), \qquad (8)$$

where $E_{t,i}$ represents the event points within the bounding box of object $i$, and $\theta_{\text{GNN}}$ are the GNN parameters.

Concurrently, features from the RGB image are extracted using a CNN, denoted as $g_{t,i}$. We concatenate the GNN and CNN features to form a comprehensive feature vector:

$$p_{t,i} = [o_{t,i}; g_{t,i}], \qquad (9)$$

which is then reduced in dimensionality via a fully connected layer:

$$f_{t,i} = \phi(p_{t,i}; \theta_0). \qquad (10)$$

**Spatio-Temporal Relational Learning.** To model temporal dependencies and interactions between objects, we employ Gated Recurrent Units (GRUs). For each object $i$, the bounding box features $b_{t,i}$ and the fused features $f_{t,i}$ are processed as follows:

$$h_{b,t,i} = \text{GRU}(b_{t,i}, h_{b,t-1,i}; \theta_1), \qquad (11)$$
$$h_{f,t,i} = \text{GRU}(f_{t,i}, h_{f,t-1,i}; \theta_2). \qquad (12)$$

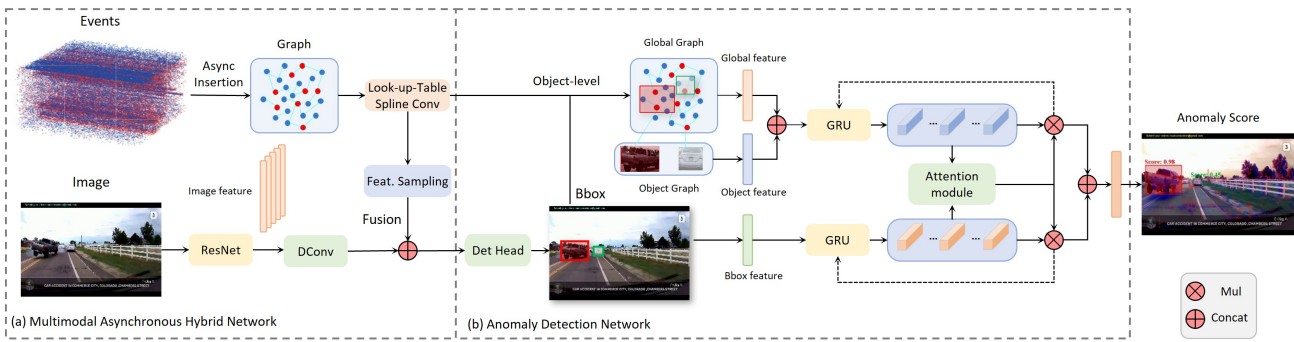

*Figure 2.* Overview of the proposed multimodal asynchronous hybrid network. (a) The framework integrates RGB images and event streams as inputs. Appearance features are extracted from RGB images using a ResNet architecture, and event features are derived from event streams through an asynchronous graph neural network (GNN) utilizing spline convolution. These features are then fused and processed by a detection head to generate object bounding boxes. (b) At the object level, features are refined through a global graph, leveraging bounding box priors, and temporal dependencies are captured using gated recurrent units (GRU). An attention mechanism dynamically assigns weights to detected objects, enhancing the focus and accuracy in anomaly detection by emphasizing anomalous objects.

Here, $\theta_1$ and $\theta_2$ are the parameters for the GRUs handling bounding box and fused features, respectively.

**Attention Mechanism.** To prioritize significant objects, we apply an attention mechanism to the GRU outputs. The attention weights for bounding box and fused features are computed as:

$$\alpha_{b,t} = \text{softmax}\left(\tanh\left(H_{b,t}^{\top} w_b\right)\right), \quad (13)$$

$$\hat{H}_{b,t} = H_{b,t}\alpha_{b,t}, \quad (14)$$

$$\alpha_{f,t} = \text{softmax}\left(\tanh\left(H_{f,t}^{\top} w_f\right)\right), \quad (15)$$

$$\hat{H}_{f,t} = H_{f,t}\alpha_{f,t}, \quad (16)$$

where $H_{b,t}$ and $H_{f,t}$ are the hidden states for bounding box and fused features, and $w_b$, $w_f$ are learnable parameters.

**Risk Score Prediction.** The attention-weighted features are concatenated to form a unified representation for each object:

$$\hat{h}_{t,i} = [\hat{h}_{b,t,i}; \hat{h}_{f,t,i}], \quad (17)$$

which is then passed through a fully connected layer and softmax activation to compute the riskiness score $s_{t,i}$:

$$s_{t,i} = \text{softmax}\left(\phi\left(\hat{h}_{t,i}; \theta_3\right)\right). \quad (18)$$

Here, $\theta_3$ are the parameters of the final classification layer.

This network architecture ensures that anomalies are detected accurately and promptly by integrating spatial features from RGB images with temporal dynamics from event streams, all processed through efficient asynchronous and recurrent mechanisms tailored for real-time performance.

## 5. Experiments

### 5.1. Experimental Setup

**Datasets.** We employ two datasets, ROL (Karim et al., 2023) and DoTA (Yao et al., 2022), both annotated with detailed temporal, spatial, and categorical information, making them highly suitable for traffic anomaly detection research. The ROL dataset provides comprehensive annotations for each video clip, which encompass object descriptions, accident details, and scene contexts. Temporal annotations pinpoint the initial appearance of a risky traffic agent and the onset of an accident, offering insights into the dynamics of risk development and collision occurrence. Spatially, the dataset includes bounding boxes for each traffic agent, which are initially detected using YOLOv5, subsequently tracked across frames with DeepSort (Veeramani et al., 2018), and finally refined by human annotators to ensure high accuracy. Categorical annotations in the dataset include traffic agent types and scene contexts, enhancing the dataset's utility for in-depth traffic behavior analysis and research. DoTA stands as the first openly available dataset tailored for traffic video anomaly detection, featuring robust temporal, spatial, and categorical annotations. It comprises over 4,600 video clips, collected under a variety of regional, weather, and lighting conditions, with each video documenting one specific anomaly. Temporal annotations in DoTA detail the start, duration, and conclusion of anomalies, while spatial annotations provide bounding boxes coupled with unique tracking IDs for each object involved in the anomalies. Currently, our DoTA dataset, ROL dataset, and all existing real-world (non-synthetic) first-person autonomous driving anomaly detection datasets lack the event modality. To address this, we utilized the v2e (Hu et al., 2021b) conversion technique to generate and supplement event modality data. This allows us to simulate the continuous event streams that would

be captured by event cameras in real-world scenarios. It is important to note that v2e is solely used for generating supplementary event data and serves no other purpose in our work.

**Implementation Details.** Our proposed multimodal anomaly detection model is implemented in PyTorch. For object detection, RGB frames are resized to $224 \times 224$ pixels and processed using a ResNet50 backbone for feature extraction. We adopt the YOLOX framework for bounding box detection, optimizing with IoU loss, class loss, and regression loss. For anomaly detection, features from the tracked objects are fed into a GRU network to capture temporal dependencies, complemented by an attention mechanism that focuses on potential anomalies. Asynchronous event data are handled using an asynchronous GNN layer, which models interactions at the frame level with bounding box priors aiding spatial-temporal analysis. The integration of spatial and temporal features from the ResNet backbone facilitates comprehensive anomaly detection.

To address cross-modality training schedule ambiguity, both modalities are trained with clearly defined parameters. The RGB component is trained for 30 epochs using a batch size of 64, and the dataset contains 1,920 images per epoch, resulting in a total of 57,600 data passes during training. For the event-based component, derived from the v2e conversion of the image-based dataset, we employ a batch size of 32 and a dataset size of 1,920 samples per epoch, with training spanning 150,000 iterations. This is equivalent to approximately 2,500 passes over the event dataset, ensuring comprehensive learning of the event-derived features.

Training employs the Adam optimizer for the GRU-attention module with a learning rate of $0.001$, and the AdamW optimizer for the GNN-ResNet combination with a learning rate of $2 \times 10^{-4}$. A ReduceLROnPlateau scheduler adjusts learning rates to optimize training stability. Class weights in the ROL dataset are set to 0.27 for the negative class and 1 for the positive class to balance the model's response. These detailed training schedules and parameters ensure effective integration of detection and attention mechanisms for both spatial and temporal anomaly identification.

## 5.2. Evaluation Metrics

To comprehensively evaluate our real-time anomaly detection model, we employ a set of metrics designed to measure both predictive accuracy and timeliness:

**Area Under the Curve (AUC).** AUC (Hanley & McNeil, 1982) quantifies how well the model distinguishes between risky and non-risky agents by computing the area under the ROC curve, thereby providing a single metric that balances true positive rate (TPR) and false positive rate (FPR).

**Average Precision (AP).** AP (Everingham et al., 2010) calculates the area under the Precision-Recall curve, offering an intuitive measure for imbalanced datasets by emphasizing both precision and recall.

**Mean Average Precision (mAP).** mAP (Lin et al., 2014) evaluates detection performance across varying IoU thresholds (from 0.5 to 0.95 in increments of 0.05), thus capturing a more nuanced perspective on overall detection robustness.

**Mean Time-to-Accident (mTTA).** mTTA (Fang et al., 2023) measures the average earliest time at which a risky agent's score $s_{t,i}$ surpasses a threshold $\bar{s}$, reflecting the model's capability to foresee accidents before they occur.

**Frame-Level AUC (AUC-Frame).** AUC-Frame evaluates the model's ability to detect risky frames within a video. It is defined as the area under the ROC curve at the frame level:

$$\text{AUC-Frame} = \int_0^1 \text{TPR}(t)\,d(\text{FPR}(t)), \qquad (19)$$

where $\text{TPR}(t)$ and $\text{FPR}(t)$ denote the true positive rate and false positive rate at threshold $t$, respectively.

**Mean Response (mResponse).** To capture not just *if* but also *how quickly* anomalies are detected at various sensitivity levels, we introduce mResponse. Unlike a single-threshold evaluation, mResponse measures the average detection delay across multiple thresholds, offering a more holistic view of real-time performance. Formally, it is defined as:

$$\text{mResponse} = \frac{1}{n}\sum_{j=1}^{n}\text{Response}_j, \qquad (20)$$

where $n$ is the number of thresholds and $\text{Response}_j$ is the detection delay at the $j$-th threshold. By aggregating response times across varying operational sensitivities, mResponse provides a more robust measure of how promptly the model flags anomalies, making it particularly suitable for real-world, safety-critical scenarios.

## 5.3. Result Analysis

We evaluated the effectiveness of our proposed model (OURS) against several established methods for anomaly detection on the ROL dataset. Performance was assessed using key metrics including Area Under the Curve (AUC), Average Precision (AP), Frame-Level AUC (AUC-Frame), and mean Time-to-Accident (mTTA). The compared models include ConvAE (Hasan et al., 2016), ConvLSTMAE (Chong & Tay, 2017), AnoPred (Liu et al., 2018), FOL (Yao et al., 2019) (with variants FOL-IoU, FOL-Mask, FOL-STD, and FOL-Ensemble (Yao et al., 2022)), MAMTCF (Liang et al., 2023), AM-Net (Karim et al., 2023), STFE (Zhou et al., 2022), and TTHF (Liang et al., 2024).

*Table 1.* Comparison of the proposed model with existing methods on the ROL and DoTA test datasets

| METHOD | AUC(%)↑ | | AP(%)↑ | | AUC-FRAME(%)↑ | | MTTA(S)↑ | | FPS↑ | MRESPONSE(S)↓ | |
|---|---|---|---|---|---|---|---|---|---|---|---|
| DATASETS | ROL | DoTA | ROL | DoTA | ROL | DoTA | ROL | DoTA | ALL | ROL | DoTA |
| CONVAE(HASAN ET AL., 2016) | 0.759 | 0.779 | 0.493 | 0.521 | 0.608 | 0.663 | 1.64 | 1.75 | 82 | 2.44 | 2.31 |
| CONVLSTMAE(CHONG & TAY, 2017) | 0.713 | 0.501 | 0.479 | 0.626 | 0.594 | 0.595 | 1.47 | 1.82 | 65 | 2.67 | 2.58 |
| ANOPRED(LIU ET AL., 2018) | 0.773 | 0.790 | 0.517 | 0.541 | 0.610 | 0.675 | 1.74 | 1.89 | 67 | 2.39 | 2.26 |
| FOL-IOU(YAO ET AL., 2022) | 0.817 | 0.830 | 0.539 | 0.563 | 0.660 | 0.730 | 1.95 | 2.09 | 56 | 2.14 | 1.79 |
| FOL-MASK(YAO ET AL., 2022) | 0.826 | 0.846 | 0.546 | 0.569 | 0.674 | 0.725 | 1.98 | 1.99 | 44 | 1.98 | 2.05 |
| FOL-STD(YAO ET AL., 2022) | 0.837 | 0.852 | 0.550 | 0.573 | 0.679 | 0.714 | 1.94 | 2.10 | 41 | 2.01 | 2.03 |
| FOL-ENSEMBLE(YAO ET AL., 2022) | 0.849 | 0.866 | 0.563 | 0.577 | 0.698 | 0.744 | 2.05 | 2.13 | 33 | 2.16 | 2.35 |
| MAMTCF(LIANG ET AL., 2023) | 0.841 | 0.862 | 0.568 | 0.581 | 0.701 | 0.766 | 2.01 | 2.11 | 98 | 1.88 | 1.81 |
| AM-NET(KARIM ET AL., 2023) | 0.855 | 0.874 | 0.576 | 0.586 | 0.707 | 0.765 | 2.18 | 2.24 | 61 | 1.96 | 1.88 |
| STFE(ZHOU ET AL., 2022) | 0.862 | 0.881 | 0.579 | 0.602 | 0.728 | 0.793 | 2.23 | 2.34 | 77 | 2.04 | 1.99 |
| TTHF(LIANG ET AL., 2024) | 0.871 | 0.891 | **0.585** | 0.618 | 0.733 | **0.847** | 2.35 | 2.41 | 3 | 2.46 | 2.58 |
| OURS | **0.879** | **0.896** | 0.570 | **0.623** | **0.736** | 0.823 | **2.80** | **2.78** | 579 | **1.17** | **1.21** |

As summarized in Table 1, OURS achieves leading performance in both AUC and AP metrics, demonstrating superior capability in distinguishing risky agents and balancing precision with recall. While the TTHF method outperforms OURS in the AUC-Frame metric, OURS significantly surpasses all other models, including TTHF, in terms of response time. This superior response time is attributed to OURS's reliance on asynchronous Graph Neural Networks (GNNs) and event stream integration, which enable an inference speed approaching 600 FPS. Additionally, OURS achieves an exceptionally low mean response time (mResponse), highlighting its promptness in detecting anomalies. This low latency is a result of the model's high inference speed and efficient anomaly detection mechanisms, as further evidenced by the favorable mTTA values. In contrast, the TTHF method, which integrates text information fusion, exhibits slower response times despite higher detection performance. Overall, the comparative analysis underscores that OURS not only sets a new benchmark in AUC and frame-level anomaly detection but also significantly enhances early risk localization capabilities. These results establish OURS as the leading model in real-time anomaly detection, combining high accuracy with exceptional timeliness, as detailed in Table 1.

### 5.4. Ablation Studies

We conducted ablation experiments to investigate the impact of various modules on model performance, focusing on metrics including AUC, AP, AUC-Frame, mTTA, and mAP. Table 2 provides a summary of the results on the ROL dataset.

**RGB + Event.** Incorporating both RGB and event data significantly enhances the model's overall performance. RGB features offer rich visual information that aids in distinguishing between object categories such as vehicles and pedestrians, while also complementing event stream data to improve object detection accuracy, as reflected by increased mAP in

diverse driving scenarios. Meanwhile, event features capture asynchronous and dynamic motion cues, effectively representing the relative movement between objects and the autonomous vehicle. This enables the model to rapidly and reliably detect anomalies, particularly in challenging conditions such as extreme lighting or at night, further strengthening the robustness of the system.

**GRU Module for Temporal Dynamics.** The GRU module is critical for capturing and leveraging temporal information. Integrating GRUs increases AUC from 0.805 to 0.817 and AP from 0.479 to 0.508, indicating a more accurate classification of anomalies. Moreover, mTTA improves from 1.44 to 1.98 seconds, demonstrating the GRU's effectiveness in accumulating temporal features and enabling early detection—an essential feature for real-time anomaly detection.

**Attention Module.** Although secondary to GRUs in modeling temporal dependencies, the Attention module considerably boosts the model's sensitivity to anomalies by concentrating on salient regions. This targeted approach proves especially beneficial in complex or cluttered environments, where focusing on relevant features is critical for accurate anomaly detection.

**BBox and Object Modules for Precise Localization.** The BBox module refines the model's localization capabilities, delivering more precise bounding box information and thereby improving AUC and AP metrics. Building on this, the Object module leverages these bounding boxes to further enhance detection accuracy by extracting detailed object-level features. This enhancement is particularly valuable in crowded or occluded scenes, where precise object recognition is challenging.

Together, the multimodal integration (RGB + Event), GRU, Attention, BBox, and Object modules comprehensively elevate the model's performance across all metrics. While RGB images increase mAP through richer spatial details, the GRU and Attention modules substantially enhance temporal

*Table 2.* Ablation study results for different model configurations on ROL dataset, showing the effects of different components on AUC, AP, AUC-Frame, mTTA, and mAP.

| RGB | EVENT | GRU | ATTENTION | BBOX | OBJECT | TWO-STAGE | AUC(%)↑ | AP(%)↑ | AUC-FRAME(%)↑ | mTTA(s)↑ | MAP(%)↑ |
|---|---|---|---|---|---|---|---|---|---|---|---|
| ✓ | ✓ | | | | | | 0.805 | 0.479 | 0.648 | 1.44 | 41.66 |
| ✓ | ✓ | ✓ | | ✓ | ✓ | | 0.817 | 0.508 | 0.668 | 1.98 | 43.59 |
| ✓ | ✓ | | ✓ | ✓ | ✓ | | 0.819 | 0.498 | 0.657 | 1.52 | 43.77 |
| | ✓ | ✓ | ✓ | ✓ | ✓ | | 0.823 | 0.518 | 0.691 | 2.06 | 35.76 |
| ✓ | ✓ | ✓ | ✓ | | | | 0.813 | 0.507 | 0.688 | 1.73 | 43.57 |
| ✓ | ✓ | ✓ | ✓ | ✓ | | | 0.839 | 0.531 | 0.703 | 2.11 | 43.29 |
| ✓ | | ✓ | ✓ | ✓ | ✓ | | 0.845 | 0.539 | 0.694 | 1.96 | 42.94 |
| ✓ | ✓ | ✓ | ✓ | ✓ | | ✓ | 0.868 | 0.561 | 0.726 | 2.51 | 43.82 |
| ✓ | ✓ | ✓ | ✓ | ✓ | ✓ | | **0.879** | **0.570** | **0.736** | **2.80** | **45.15** |

detection accuracy and sensitivity. Simultaneously, BBox and Object modules refine localization and object recognition, culminating in a robust, high-performing anomaly detection framework.

Our multimodal asynchronous hybrid network is designed in a modular fashion, allowing the network depth (i.e., number of ResBlocks and look-up-table Spline convolution layers) to be increased for more complex data. Experiments show that increasing the number of layers slightly improves detection accuracy, with a modest increase in inference latency. See Table 3.

*Table 3.* Performance and latency trade-off with different network depths on ROL.

| LAYERS | AUC | AP | AUC-F | mTTA | FPS | mRESPONSE |
|---|---|---|---|---|---|---|
| 4 | 0.879 | 0.570 | 0.736 | 2.80 | 579 | 1.17 |
| 5 | 0.885 | 0.574 | 0.739 | 2.89 | 312 | 1.31 |
| 6 | 0.892 | 0.577 | 0.740 | 2.93 | 166 | 1.56 |

These results indicate that our model is highly scalable, with only minor latency trade-offs for improved detection accuracy in increasingly complex environments.

To further enhance global feature modeling, we replaced the original CNN backbone with ViT and Swin Transformer. Transformer architectures can capture long-range dependencies and improve detection accuracy. However, the self-attention mechanism has $\mathcal{O}(N^2)$ complexity, introducing additional inference latency. Experimental results are shown in Table 4.

*Table 4.* Performance comparison of different backbone architectures on ROL.

| MODEL | AUC | AP | AUC-F | mTTA | FPS | mRESPONSE |
|---|---|---|---|---|---|---|
| OURS(CNN) | 0.879 | 0.570 | 0.736 | 2.80 | 579 | 1.17 |
| CNN→SWIN | 0.881 | 0.576 | 0.739 | 2.85 | 278 | 1.44 |
| CNN→ViT-B | 0.886 | 0.581 | 0.745 | 2.87 | 213 | 1.51 |

These results show that Transformer backbones can improve detection accuracy when latency is carefully managed, but a trade-off exists between accuracy and real-time requirements.

**5.5. Qualitative Evaluation**

Our model leverages multiple feature types: RGB features encode appearance, bounding box features represent object position and movement, event stream features capture rapid and abnormal motion, and object-level features describe local details. The object-level attention mechanism assigns scores to detected objects, highlighting those most relevant to anomaly detection.

As illustrated in Figure 4, when a vehicle suddenly cuts in, its attention score increases as it approaches, indicating its growing importance as a potential anomaly. Visualizations of these attention scores across frames demonstrate how the model dynamically focuses on critical objects, providing insight into which features are most vital for identifying anomalies. This enhanced interpretability aids in understanding the model's decision-making process and improves its credibility and deployment safety.

Figure 3 presents three challenging scenarios from the ROL dataset that highlight our model's effectiveness in detecting traffic anomalies critical for autonomous vehicle safety. These examples involve small objects and complex abnormal movements. Each example includes three rows. The top row shows Ground Truth, where normal objects appear in white and anomalies in red. The middle row displays our model's predictions, highlighting anomalies with scores above 0.5 in red. The bottom row compares frame-level anomaly scores to the Ground Truth, offering a straightforward visual measure of our model's accuracy.

**Lane Merger Scenario.** In Figure 3(a), a vehicle abruptly merges into the autonomous vehicle's lane. At frame 20, the anomaly score is 0.33, signifying the early onset of abnormal behavior. By frame 30, the score rises to 0.58, accurately highlighting the vehicle's unusual trajectory and intensified event flow. Here, the GRU module proves essential by aggregating temporal cues, boosting the detection sensitivity as the threat unfolds.

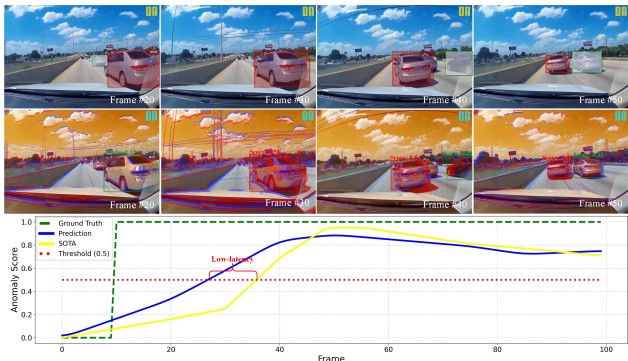
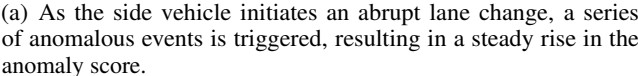

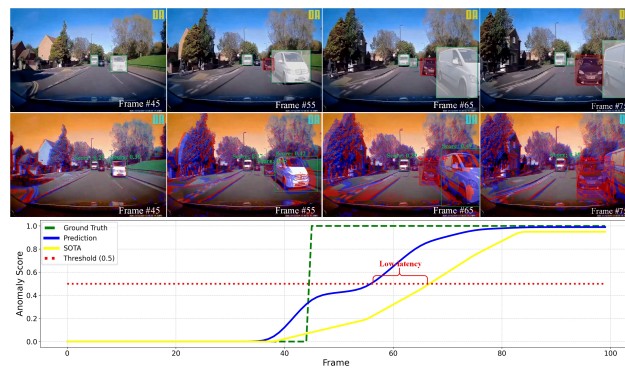

(a) As the side vehicle initiates an abrupt lane change, a series of anomalous events is triggered, resulting in a steady rise in the anomaly score.

(b) A vehicle abruptly enters the autonomous vehicle's field of view with an intercepting trajectory, causing a sharp spike in the anomaly score.

*Figure 3.* In various traffic anomaly scenarios, the actions of hazardous vehicles significantly threaten the autonomous vehicle, causing elevated anomaly scores.

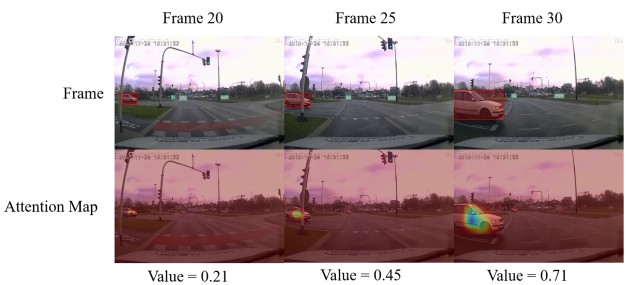

*Figure 4.* When a vehicle suddenly cuts in, its attention score increases as it approaches, indicating its growing importance as a potential anomaly.

fully emerge in the scene. This capability significantly lowers detection latency, which can be crucial for avoiding potential collisions.

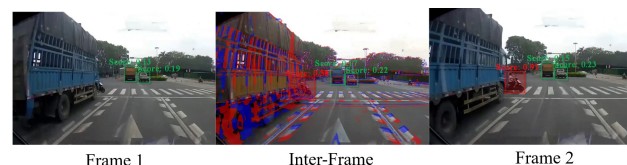

*Figure 5.* In scenarios where high-speed objects suddenly emerge, a continuous stream of events helps the model perform inter-frame anomaly detection, allowing for earlier and more timely anomaly detection.

**Oncoming Collision Risk.** Figure 3(b) depicts an oncoming vehicle veering toward the autonomous vehicle, posing a high collision risk. The model's anomaly score quickly escalates, driven by erratic movement and close proximity, which are detected by leveraging bounding box data to assess relative positions. This underscores the importance of precise spatial cues for robust anomaly detection.

**Early Detection Advantage.** Comparisons in Figure 3 indicate that our method achieves earlier anomaly detection than AM-Net (Karim et al., 2023), thereby reducing latency in critical situations. These examples confirm that our model not only covers a wide spectrum of anomalies but also reacts with temporal precision essential for real-world autonomous driving applications.

**Inter-Frame Anomaly Detection.** Figure 5 illustrates a sudden pedestrian appearance across two consecutive frames. Harnessing the asynchronous event stream properties allows the model to detect anomalies between frames, effectively anticipating the presence of fast-moving objects before they

## 6. Conclusion

We focus on the task of real-time anomaly detection in autonomous driving, underscoring the need to minimize response time without compromising detection accuracy. To address this challenge, we proposed a multimodal asynchronous hybrid network that combines event streams from event cameras with RGB image data. By employing an asynchronous GNN for high-temporal-resolution event data and a CNN for rich spatial features, our framework captures both the temporal dynamics and spatial details of driving environments. Extensive experiments on benchmark datasets demonstrate that our approach significantly surpasses existing methods in detection accuracy and response time, achieving millisecond-level responsiveness. This work lays a foundation for future research in time-critical perception tasks and advances safe, reliable deployment of autonomous vehicles.

## Impact Statement

This paper introduces a multimodal asynchronous hybrid network for real-time anomaly detection in autonomous driving. By integrating sparse event stream data with conventional image-based inputs, the method reduces detection delays, enhances response times, and improves overall safety and reliability in dynamic driving environments. Its broader impact lies in potentially reducing traffic accidents and fatalities, but it also raises ethical concerns, including job displacement, potential system failures, and privacy issues—particularly in urban settings where both public and private data may be collected. Ensuring responsible deployment requires rigorous testing, robust safety measures, and appropriate regulatory frameworks that uphold transparency, fairness, and accountability. Future work should address these challenges and any unintended social consequences, enabling autonomous driving technology to realize its transformative potential safely and ethically.

## Acknowledgements

The authors gratefully acknowledge financial support from the National Natural Science Foundation of China (Grant Nos. 62332002, 62027804, 61825101, and 62402015), the China Postdoctoral Science Foundation (Grant No. 2024M750100), and the Postdoctoral Fellowship Program of the China Postdoctoral Science Foundation (Grant No. GZB20230024). Computational resources were kindly provided by Pengcheng Cloudbrain.

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

## A. Ablation Studies

*Table 5.* Ablation study results for different model configurations on ROL dataset, showing the effects of different components on AUC, AP, AUC-Frame, mTTA, and mAP.

| RGB | Event | GRU | Attention | Bbox | Object | Two-stage | AUC(%)↑ | AP(%)↑ | AUC-Frame(%)↑ | mTTA(s)↑ | mAP(%)↑ |
|-----|-------|-----|-----------|------|--------|-----------|---------|--------|---------------|----------|---------|
| ✓ | ✓ | | | | | | 0.805 | 0.479 | 0.648 | 1.44 | 41.66 |
| ✓ | ✓ | ✓ | | ✓ | ✓ | | 0.817 | 0.508 | 0.668 | 1.98 | 43.59 |
| ✓ | ✓ | | ✓ | ✓ | ✓ | | 0.819 | 0.498 | 0.657 | 1.52 | 43.77 |
| | ✓ | ✓ | ✓ | ✓ | ✓ | | 0.823 | 0.518 | 0.691 | 2.06 | 35.76 |
| ✓ | ✓ | ✓ | ✓ | | | | 0.813 | 0.507 | 0.688 | 1.73 | 43.57 |
| ✓ | ✓ | ✓ | ✓ | ✓ | | | 0.839 | 0.531 | 0.703 | 2.11 | 43.29 |
| ✓ | | ✓ | ✓ | ✓ | ✓ | | 0.845 | 0.539 | 0.694 | 1.96 | 42.94 |
| ✓ | ✓ | ✓ | ✓ | ✓ | | ✓ | 0.868 | 0.561 | 0.726 | 2.51 | 43.82 |
| ✓ | ✓ | ✓ | ✓ | ✓ | ✓ | | **0.879** | **0.570** | **0.736** | **2.80** | **45.15** |

We conduct a comprehensive ablation study presented in Table 5, with the analysis detailed as follows:

**RGB and Event Modalities.** Both RGB and Event modalities are crucial for the model's performance. RGB features capture the static appearance of objects, while Event features extract dynamic motion and anomaly-related characteristics. As illustrated in Figure 6, objects exhibiting abrupt and irregular motions generate a significant number of anomalous events. In the fourth and seventh groups of the ablation study, the exclusion of RGB and Event inputs, respectively, led to performance degradation. Specifically, omitting RGB features caused a substantial decline in anomaly detection accuracy and mean Average Precision (mAP) for object detection, as object detection heavily relies on appearance information. Similarly, removing Event inputs reduced anomaly detection accuracy and the Time to Anomaly (TTA) metric, since the Event stream provides essential inter-frame information that enables the model to detect anomalies more promptly. These results highlight the complementary nature of the two modalities: RGB features provide rich static appearance information, while Event streams capture dynamic motion information, thereby enhancing the sensitivity of anomaly detection.

**Two-Stage vs. Single-Stage Architecture.** The eighth and ninth experiments compare the model's performance using a two-stage versus a single-stage architecture. In the two-stage architecture, Event and RGB features are first utilized for object detection, and the resulting detections serve as priors for anomaly detection. In contrast, the single-stage architecture directly leverages fused Event and RGB features for both tasks simultaneously. The superior performance of the single-stage model can be attributed to its shared feature extraction module, which allows anomaly detection to benefit directly from features optimized for object detection, thereby enhancing sensitivity to anomalous behaviors.

**Bounding Box (BBox) Priors.** The fifth and sixth experiments investigate the impact of using BBox priors for anomaly detection. BBox priors provide critical information about the relative positions of objects concerning the autonomous vehicle, which is particularly valuable for detecting anomalies. As shown in Figure 8(b), irregular BBox movements often serve as early indicators of anomalies. For instance, when a vehicle abruptly merges into the lane of the autonomous vehicle, this irregularity can be anticipated from the temporal movement of the BBox. The absence of BBox priors led to decreased anomaly detection accuracy. Unlike the DoTA (Yao et al., 2022) framework, which bases anomaly detection on predicting future object trajectories and focuses on irregular BBox movements, our method employs a Gated Recurrent Unit (GRU) to capture the temporal sequence of BBox movements, leveraging this information for anomaly detection. The inclusion of BBox priors enhances anomaly detection by providing essential positional information, and their absence negatively impacts performance, underscoring their significance.

**GRU and Attention Modules.** The second and third experiments assess the importance of the GRU and attention modules within the anomaly detection framework. Compared to the ninth experiment, removing either the GRU or the attention module resulted in a significant decrease in anomaly detection accuracy. The GRU is pivotal for modeling temporal information, as most anomalies in the ROL and DoTA datasets develop over time. Anomalies are often characterized by gradually accumulating abnormal features rather than sudden appearances. For example, as depicted in Figure 9(c), a merging vehicle requires approximately 20 frames to transition into the lane of the autonomous vehicle. During this period, the GRU incrementally aggregates anomaly features, resulting in a continuously increasing anomaly score. This demonstrates the GRU's role in modeling the temporal accumulation of anomaly features.

While the GRU captures temporal anomaly accumulation, the attention module focuses on spatial anomalies at each time step. It assigns higher attention weights to anomalous objects at the object level, enabling the model to prioritize these anomalies

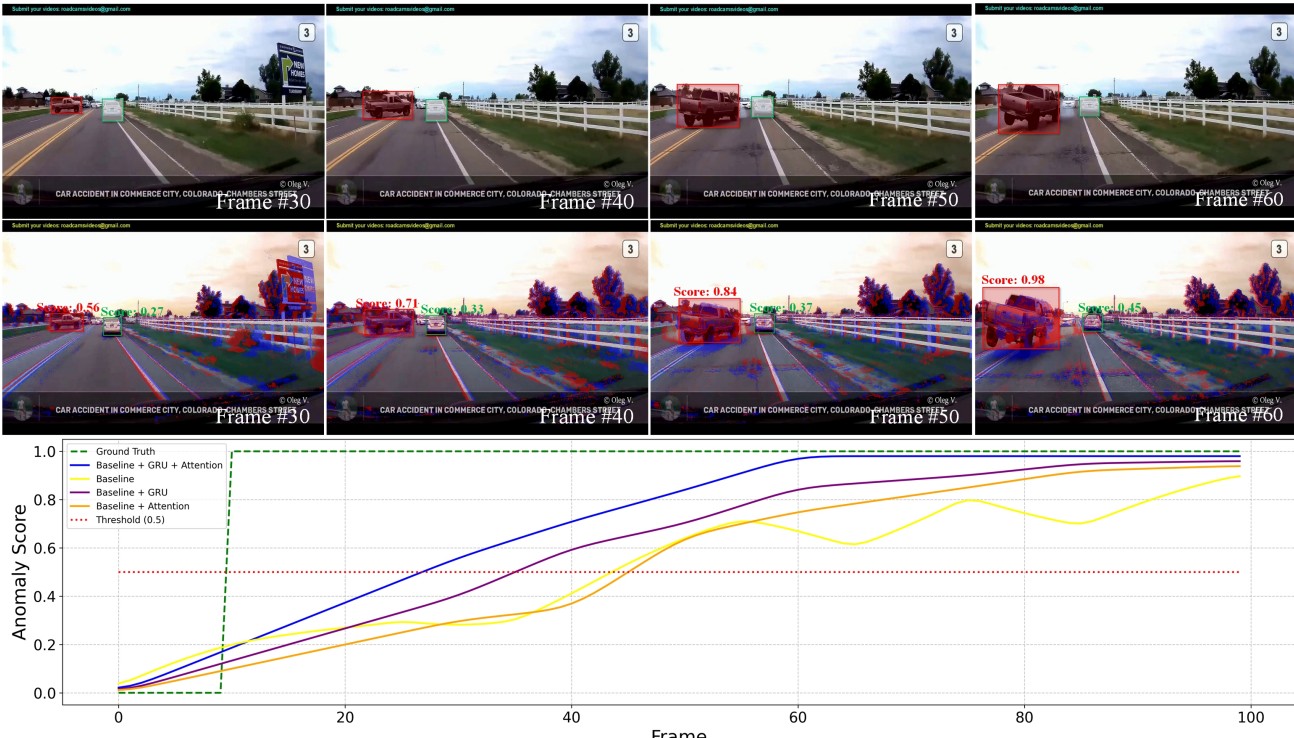

*Figure 6.* In scenarios where high-speed objects suddenly emerge, a continuous stream of events helps the model perform inter-frame anomaly detection, allowing for earlier and more timely anomaly detection.

and assign them higher anomaly scores. Together, the GRU and attention modules provide complementary strengths: the GRU tracks temporal patterns, and the attention module enhances the spatial prioritization of anomalous objects. Ablation studies show that incorporating the attention module significantly improves anomaly detection performance, confirming its critical role in enhancing detection accuracy. Figure 6 illustrates the interaction between the GRU and attention modules. The GRU facilitates faster accumulation of anomaly scores, allowing for earlier detection, while the attention module enables more accurate anomaly judgments. In contrast, the baseline anomaly score curve without GRU and attention exhibits greater fluctuations.

Overall, the ablation analysis underscores the importance of multimodal inputs (RGB and Event), BBox priors, and temporal-spatial mechanisms (GRU and attention) for effective anomaly detection. These findings validate that each component plays a vital role in enhancing the model's performance, particularly in dynamic environments where both temporal and spatial anomaly features are essential.

*Table 6.* Ablation study results for different two-stage model configurations on ROL dataset, showing the effects of different components on AUC, AP, AUC-Frame, mTTA.

| No. | RGB | BBox | Event(asyn.) | Event(sync.) | Flow | GRU | Attention | AUC(%)↑ | AP(%)↑ | AUC-Frame(%)↑ | mTTA(s)↑ |
|---|---|---|---|---|---|---|---|---|---|---|---|
| 1 | | ✓ | | | ✓ | ✓ | | 0.855 | 0.559 | 0.702 | 2.18 |
| 2 | | ✓ | | ✓ | | ✓ | | 0.840 | 0.544 | 0.698 | 2.25 |
| 3 | | ✓ | ✓ | | | ✓ | | 0.851 | 0.542 | 0.705 | 2.50 |
| 4 | | ✓ | ✓ | | | ✓ | ✓ | 0.862 | 0.559 | 0.713 | 2.48 |
| 5 | ✓ | ✓ | ✓ | | | ✓ | ✓ | **0.868** | **0.561** | **0.726** | **2.51** |

## B. Two-Stage versus Single-Stage Networks

The primary difference between two-stage and single-stage networks lies in their operational flow. Two-stage networks initially perform object detection and subsequently conduct anomaly detection without reusing the features extracted during

the object detection phase. Table 6 summarizes the impact of various components on the performance of the two-stage model.

**Experiment 1.** In the first experiment, only bounding box (BBox) features and optical flow were utilized as input modalities, with a Gated Recurrent Unit (GRU) module responsible for temporal feature extraction. This configuration achieved a high anomaly detection score but exhibited a lower mean Time to Anomaly (mTTA) due to the synchronous nature of optical flow. These results indicate that while optical flow effectively captures motion-related features, its performance is constrained by its reliance on synchronized data.

**Experiments 2 and 3.** In the second and third experiments, optical flow was replaced with synchronous and asynchronous event streams, respectively. Synchronous event streams were generated by aggregating events within fixed time windows into event frames, which were then processed similarly to optical flow. However, this synchronous configuration resulted in the lowest Area Under the Curve (AUC) and mTTA, as it negated the inherent asynchronous properties of event streams. Conversely, asynchronous event streams, processed using an asynchronous Graph Neural Network (GNN), significantly outperformed synchronous streams by leveraging motion-related information essential for anomaly detection. The asynchronous approach maintained the temporal granularity of raw events, which is critical for identifying subtle anomalies.

**Experiment 4.** The fourth experiment introduced an attention mechanism to focus on critical features, leading to further improvements in detection performance. The attention module enhanced the model's ability to identify anomalies by selectively weighting more relevant object-level features. This underscores the importance of prioritizing specific regions and features, especially in complex anomaly detection scenarios.

**Experiment 5.** In the fifth experiment, RGB features were incorporated alongside BBox and asynchronous event streams, resulting in the highest overall performance improvement. Although RGB features had a limited direct impact on anomaly detection, they indirectly enhanced performance by improving object detection accuracy. This demonstrates that RGB features, while not directly influential in anomaly detection, play a supportive role by providing more accurate object localization and detection in the initial stage.

Overall, these findings highlight the effectiveness of asynchronous event streams and attention mechanisms in enhancing two-stage networks. Additionally, they acknowledge the indirect benefits of RGB features through improved object detection, thereby contributing to more robust anomaly detection performance.

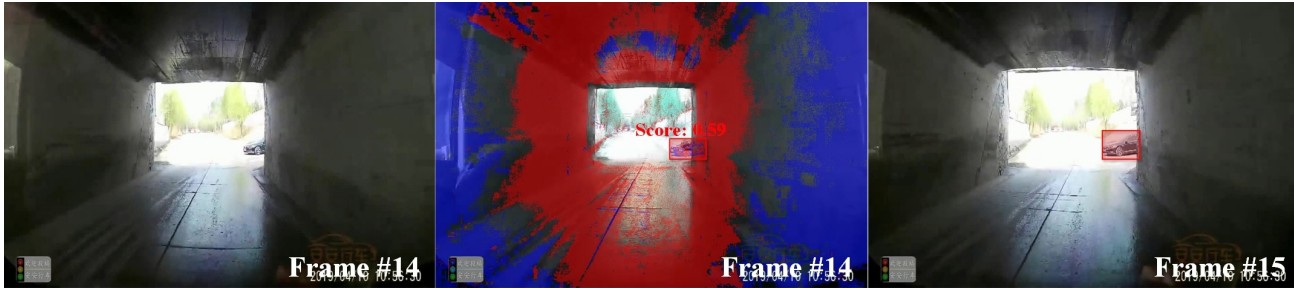

*Figure 7.* Examples from the Rush-Out dataset demonstrate the effectiveness of our approach. At a tunnel exit under intense backlighting, a vehicle suddenly emerges from the tunnel edge. Leveraging the advantages of event data, our method enables rapid and accurate anomaly detection in such challenging scenarios.

## C. New Datasets

The ROL (Karim et al., 2023) and DoTA (Yao et al., 2022) datasets encompass a diverse range of traffic anomalies, including scenarios both with and without the presence of autonomous vehicles. To specifically evaluate the low-latency capabilities of our model, we identified and selected scenarios that best highlight this aspect. These scenarios are characterized by the sudden appearance of pedestrians or vehicles, posing significant risks to autonomous vehicles. Such abrupt occurrences typically result from the occlusion by large objects or the autonomous vehicle's narrow field of view, causing fast-moving entities to unexpectedly enter the vehicle's sight and create hazardous situations. In these contexts, the low response latency of our method is particularly advantageous.

*Table 7.* Comparison of the proposed model with existing methods on the Rush-Out test dataset

| Method | AUC(%)↑ | AP(%)↑ | AUC-Frame(%)↑ | mTTA(s)↑ | mResponse(s)↓ |
|---|---|---|---|---|---|
| ConvAE(Hasan et al., 2016) | 0.789 | 0.502 | 0.612 | 1.03 | 1.88 |
| ConvLSTMAE(Chong & Tay, 2017) | 0.738 | 0.490 | 0.603 | 1.05 | 2.05 |
| AnoPred(Liu et al., 2018) | 0.790 | 0.521 | 0.616 | 1.14 | 1.92 |
| FOL-IoU(Yao et al., 2022) | 0.823 | 0.543 | 0.666 | 1.30 | 1.61 |
| FOL-Mask(Yao et al., 2022) | 0.835 | 0.553 | 0.681 | 1.35 | 1.54 |
| FOL-STD(Yao et al., 2022) | 0.845 | 0.561 | 0.689 | 1.32 | 1.48 |
| FOL-Ensemble(Yao et al., 2022) | 0.870 | 0.572 | 0.698 | 1.45 | 1.62 |
| AM-Net(Karim et al., 2023) | 0.879 | 0.579 | 0.710 | 1.48 | 1.55 |
| OURs | **0.887** | **0.592** | **0.755** | **1.71** | **1.29** |

To facilitate this evaluation, we curated the **Rush-out**[1] dataset by extracting instances of dangerous driving scenarios involving sudden outbursts of objects from the ROL and DoTA datasets. The Rush-out dataset comprises 1,084 videos, each recorded at 20 frames per second (fps) with a resolution of $1280 \times 720$. Table 7 presents a performance comparison on the Rush-out dataset using the same baseline methods referenced in Table 5. Our proposed method surpasses the state-of-the-art (SOTA) by 0.23 seconds in the mean Time to Anomaly (mTTA) metric and achieves a 0.19-second improvement in the mean Response Time (mResponse) metric.

The Rush-out dataset predominantly features high-speed scenes where even a single frame can result in significant displacement of an anomalous vehicle, thereby increasing the associated risk. Despite the heightened complexity of these scenarios, our method consistently demonstrates superior performance in anomaly detection metrics.

Figure 7 and Figure 8 illustrate the results obtained on the Rush-out dataset. Specifically, Figure 8 depicts a scenario where a vehicle abruptly emerges from the edge of a tunnel exit under strong backlighting conditions. The intense lighting induces significant visual blurring, rendering traditional detection methods less effective for fast-moving objects. Our approach leverages event streams, which are particularly robust under extreme lighting conditions, highlighting the unique advantage of event cameras in complementing conventional RGB imagery. By utilizing this modality, our method reliably detects rapidly emerging vehicles across consecutive frames, enabling early anomaly detection and timely alerts in such challenging environments.

## D. Qualitative Evaluation

Our model prioritizes rapid inference speed and minimal anomaly detection latency, which are crucial in scenarios where fast-moving objects suddenly appear. Figure 8 presents typical instances of pedestrians or vehicles emerging unexpectedly. By leveraging continuous event streams between consecutive video frames, our model performs inter-frame anomaly detection, thereby effectively reducing detection latency.

Figure 8(a) illustrates a case where a boy suddenly runs out from behind a truck. In this scenario, the boy enters the field of view mid-road due to a blind spot. Although the boy is not detected in the first frame, the continuous event stream between the first and second frames enables the model to detect the boy in advance, assigning a high anomaly score and significantly reducing detection latency.

Similarly, Figure 8(b) and Figure 8(c) depict vehicles abruptly emerging from the side. Due to their high speed, the model utilizes event streams between frames for inter-frame anomaly detection, allowing for earlier identification of anomalies and further reduction in detection latency.

Figure 9 showcases additional results of real-time traffic anomaly detection, where most cases involve abnormal vehicles or pedestrians obstructing the autonomous vehicle's path, leading to traffic anomalies.

In Figure 9(a), a turning vehicle increasingly approaches the autonomous vehicle, resulting in elevated anomaly scores. The relative position between the abnormal vehicle and the autonomous vehicle is a key factor in this detection.

---

[1]https://www.filecad.com/7xga/rush-out.zip

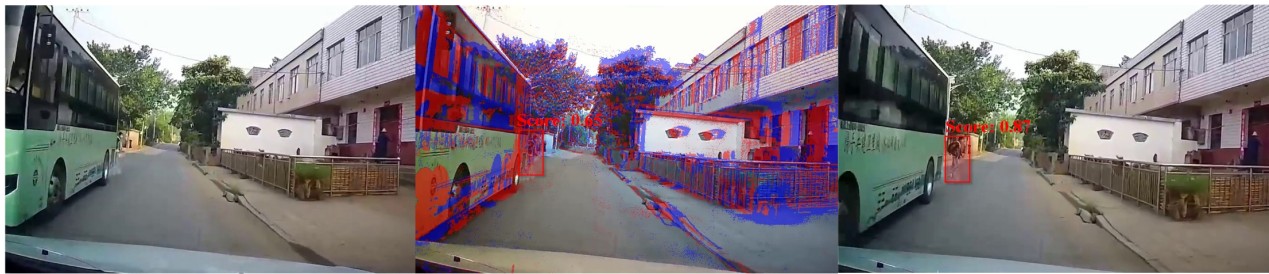

(a) The boy suddenly rushed out from behind the truck on the left.

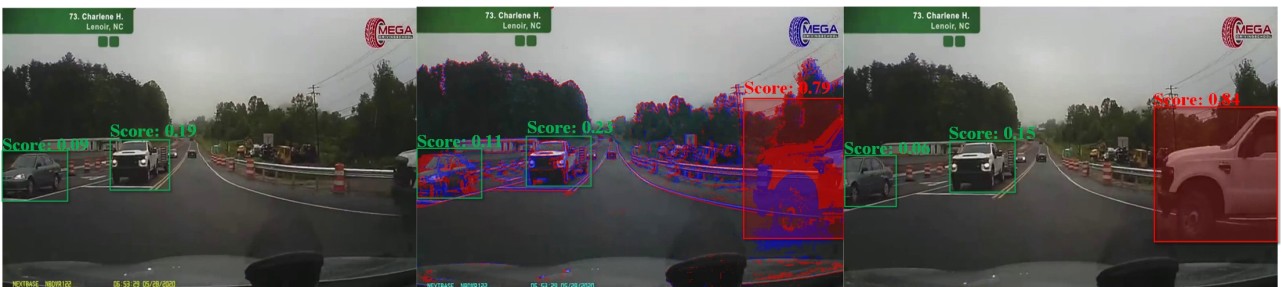

(b) A pickup truck suddenly rushed out from the right side of the field of vision.

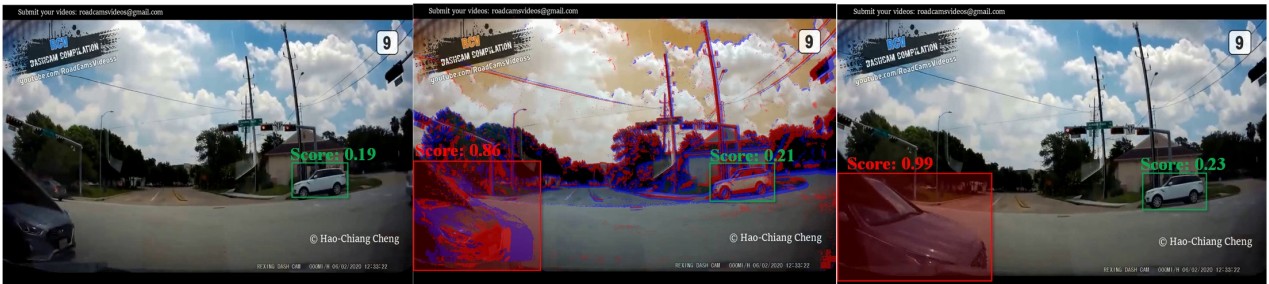

(c) A pickup truck suddenly rushed out from the right side of the field of vision.

*Figure 8.* The anomaly detection results of suddenly rushing out of the scene, the objects moving at high speed in two consecutive frames can use the characteristics of event streams to help detect abnormalities between two frames.

Figure 9(b) shows a cyclist suddenly changing lanes, creating a collision risk with the autonomous vehicle. The cyclist's irregular motion generates high anomaly scores, as these irregular movements produce numerous anomalous event streams that are central to anomaly detection.

Finally, Figure 9(c) depicts a scenario where a normally moving vehicle on the right abruptly changes lanes, posing a collision risk with the autonomous vehicle. The vehicle's irregular lane change, combined with its decreasing distance from the autonomous vehicle, is the primary factor for anomaly detection.

## E. Failure Cases and Limitations

Our system's performance relies heavily on the accuracy of the object detector. As shown in Figure 10, in Frame 25 and Frame 30, blurry images and small objects led to complete detection failure, which in turn caused the anomaly detection framework to fail. Although an object was detected in Frame 35 when it was very close to the ego vehicle, the attention score was only 0.56 due to missing previous detections, preventing the GRU from building a robust temporal feature set. These cases highlight that our approach is limited by the reliability of the object detector, and improvements in detection accuracy are essential.

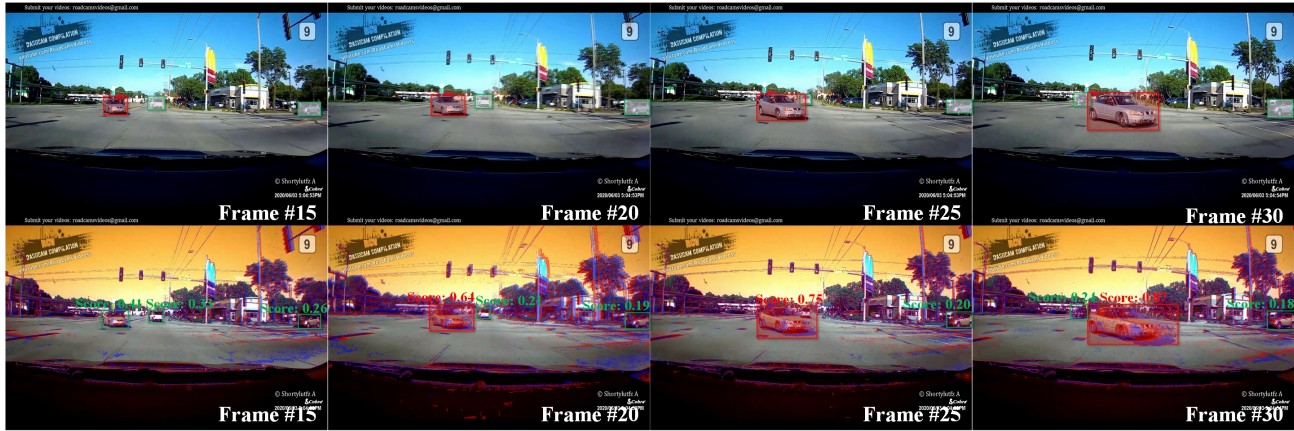

(a) The vehicle ahead on the left makes a turn, progressively reducing its distance to the autonomous vehicle, posing a significant risk.

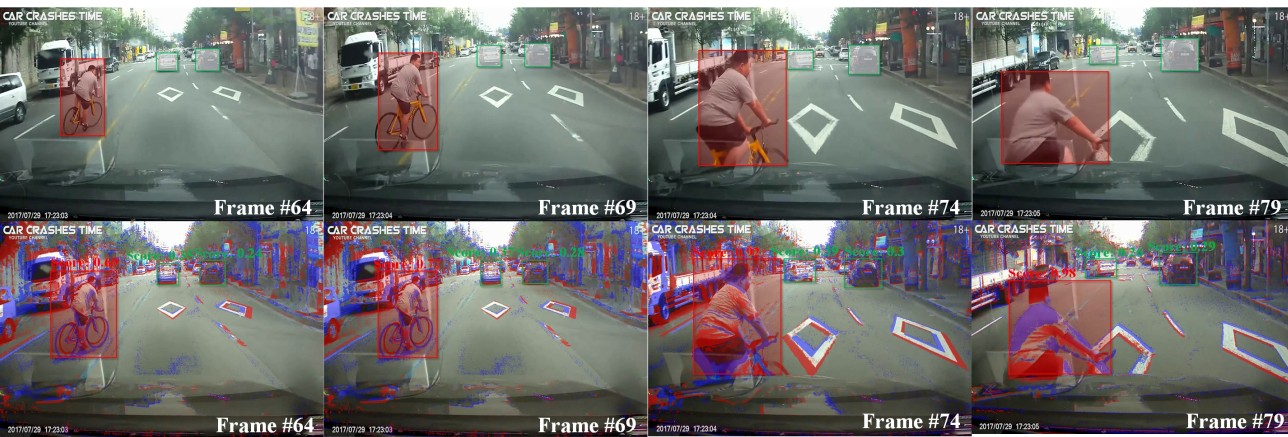

(b) A cyclist on the left changes lanes unexpectedly, creating a high risk of collision with the autonomous vehicle, which ultimately results in a crash.

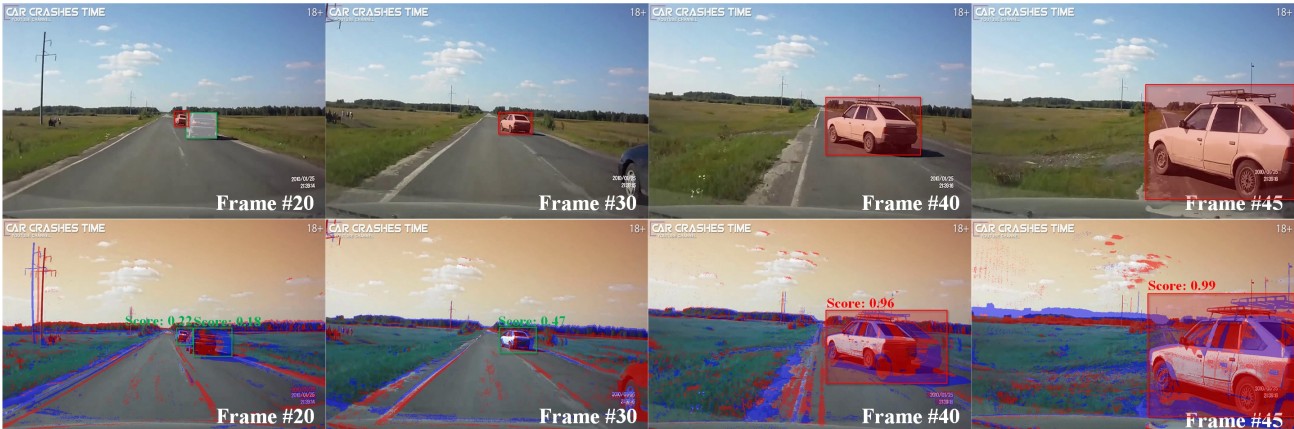

(c) A vehicle on the right, initially driving normally, suddenly changes lanes, decreasing its distance to the autonomous vehicle and leading to a high anomaly risk.

*Figure 9.* Additional traffic anomaly scenarios for real-time anomaly detection primarily involve abnormal vehicles obstructing the autonomous vehicle's path, posing significant risks to its operation.

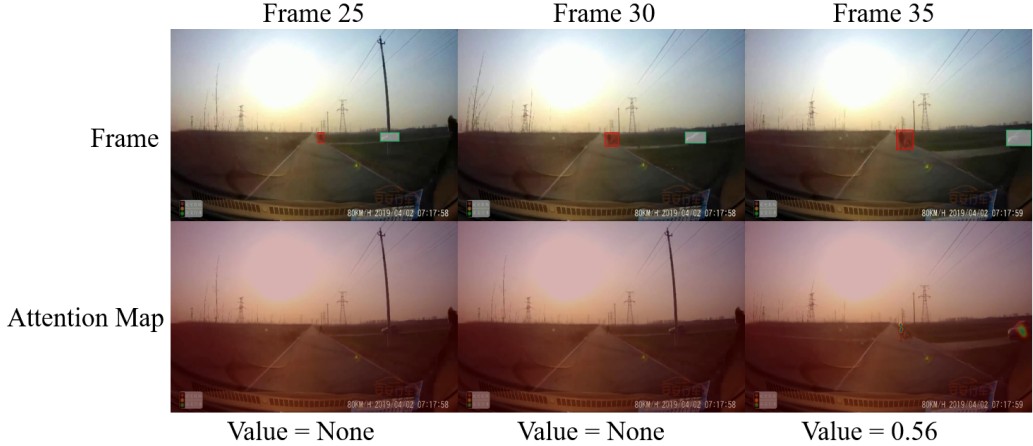

*Figure 10.* Our approach is limited by the reliability of the object detector, and improvements in detection accuracy are essential.

## F. Sensitivity Analysis

Our model remains robust across different settings. At the standard 0.5 IOU and 0.5 confidence thresholds, the model achieves an AUC of 0.879. As shown in Figure 11, increasing these thresholds sharply reduces AUC by filtering out more detection boxes, while lowering them causes a gradual decline as false positives are effectively down-weighted. Overall, the AUC ranges from 0.65 to 0.879, confirming the stability of our configuration.

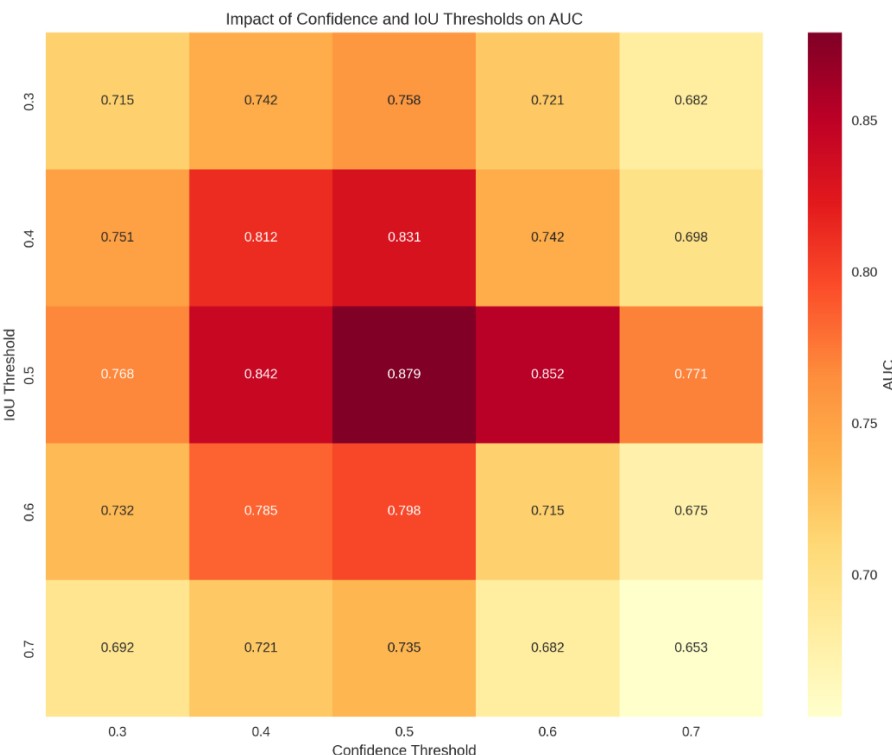

*Figure 11.* Sensitivity analysis of the model's hyperparameters IOU threshold and target detection confidence threshold.

## G. Real-Time Anomaly Detection Methods

We compared our approach with several recent real-time video anomaly detection methods, including AED-MAE (Ristea et al., 2024), EfficientAD (Batzner et al., 2024), and MOVAD (Rossi et al., 2024). AED-MAE is a state-of-the-art method for surveillance videos that emphasizes detection speed, but relies on static background modeling, making it less suitable for dynamic traffic environments. EfficientAD is designed for industrial anomaly detection and also operates at the frame level. MOVAD is the first online real-time anomaly detection method specifically for autonomous driving. Our approach achieves the best balance between accuracy and speed, meeting the unique requirements of the traffic domain. Results on ROL and DoTA are shown in Table 8.

*Table 8.* Comparison with recent real-time anomaly detection methods on ROL and DoTA.

| METHOD | TYPE | AUC-FRAME(%)↑ | | mTTA(S)↑ | | mRESPONSE(S)↓ | | FPS↑ |
|---|---|---|---|---|---|---|---|---|
| DATASETS | - | ROL | DoTA | ROL | DoTA | ROL | DoTA | - |
| EFFICIENTAD | FRAME | 0.519 | 0.549 | 0.89 | 0.97 | 3.65 | 2.68 | 557 |
| AED-MAE | FRAME | 0.571 | 0.652 | 1.01 | 1.35 | 3.36 | 2.79 | 1655 |
| MOVAD | FRAME | 0.719 | 0.821 | 2.47 | 2.55 | 2.61 | 1.33 | 158 |
| **OURS** | OBJECT | **0.736** | **0.823** | **2.80** | **2.78** | **2.35** | **1.21** | **579** |

## H. Synthetic Event Data and Validation

Currently, mainstream traffic anomaly detection datasets do not contain real event (DVS) modality. To address this, we employ the V2E (Video to Event) method to convert conventional video data into event streams, thereby supplementing the event modality input. V2E effectively simulates core DVS sensor characteristics, including Gaussian threshold distribution, temporal noise, leak events, and intensity-dependent bandwidth. Its fidelity has been validated on datasets such as N-Caltech 101, where V2E-generated data raised ResNet34 accuracy on real DVS from 81.69% to 83.36%, and up to 87.85% after fine-tuning (compared to 86.74% using real data only).

Although the DSEC (Gehrig et al., 2021) dataset contains real event data in autonomous driving scenarios, it only covers normal driving and lacks anomaly events, making it unsuitable for direct use in anomaly detection tasks. To further validate the effectiveness of V2E-generated data, we generated a simulated DSEC+V2E dataset and conducted experiments on ROL and DoTA. The results show that using DSEC+V2E versus DSEC alone leads to minimal changes in anomaly detection performance (AUC, AP, mTTA, etc.), confirming that the V2E method can effectively supplement the event modality and maintain model robustness. See Table 9.

*Table 9.* Performance comparison of DSEC and DSEC+V2E on the ROL and DoTA test sets.

| METRICS | AUC(%)↑ | | AUC-FRAME(%)↑ | | mTTA(S)↑ | | mRESPONSE(S)↓ | |
|---|---|---|---|---|---|---|---|---|
| DATASET | ROL | DoTA | ROL | DoTA | ROL | DoTA | ROL | DoTA |
| DSEC | 0.841 | 0.857 | 0.697 | 0.794 | 2.24 | 2.18 | 1.66 | 1.79 |
| DSEC+V2E | 0.846 | 0.862 | 0.712 | 0.808 | 2.46 | 2.37 | 1.45 | 1.61 |

In summary, despite inherent differences between synthetic and real event data, our systematic experiments demonstrate the effectiveness of V2E and the robustness of our detection model, laying the groundwork for future real event-based anomaly detection datasets.

## I. Deployment Feasibility and Hardware Adaptation

Our method can be deployed end-to-end on mainstream autonomous driving chips. For example, on Orin chips, the system can process approximately 560k events per second under normal load, and up to 10M events/second at peak, with a total computational cost of about 87.32 TFLOPs and 42W power consumption, which is within the chip's capability. Although two cameras (RGB + Event) are used, hardware synchronization limits timing errors to 78 microseconds, and the event camera itself introduces only about 6ms delay, meeting real-time requirements. For resource-constrained chips, quantization and other optimizations can further improve deployment efficiency.

## J. Computational Analysis

Our method requires only 8.732 MFLOPs per event and uses 23.5 GB of video memory. Under typical conditions in the ROL dataset, with an average event rate of 560k events per second, this overhead is manageable. In high-speed driving scenarios—where the event rate can rise to 1–10M events/s—the worst-case computational load is approximately 87.32 TFLOPs.

Current autonomous driving chips such as Atlan, Thor, and Orin can fully deploy our algorithm on-board, while chips like Xavier and Parker, although sufficient for normal driving, might face challenges under extreme conditions. The specifications of several representative chips are summarized in Table 10.

*Table 10.* Representative autonomous driving chip specifications.

| CHIP | ARCHITECTURE | COMPUTE POWER |
|---|---|---|
| PARKER | 2×DENVER | 1 TOPS |
| XAVIER | 4×ARM CORTEX A578 | 30 TOPS |
| ORIN | NVIDIA CUSTOM CARMEL | 254 TOPS |
| ATLAN | ARM6412 × ARM CORTEX-A78 | 1000 TOPS |
| THOR | AE (HERCULES) ARM NEOVERSE V2 | 2000 TFLOPS @ FP8 |

## K. Extreme scenarios

We collected data on severe weather and low-light scenes from the ROL and DoTA datasets, and performed comparative experiments with the two best-performing baseline methods. Benefiting from the advantages of event cameras in extreme lighting scenarios, our method outperforms other methods by a large margin in these scenarios. See Table 11.

*Table 11.* Performance comparison in adverse weather or low-light scenarios.

| METHOD | AUC(%)↑ | AP(%)↑ | AUC-FRAME(%)↑ | MTTA(S)↑ | MRESPONSE(S)↓ |
|---|---|---|---|---|---|
| STFE | 0.631 | 0.397 | 0.582 | 1.26 | 3.12 |
| TTHF | 0.652 | 0.416 | 0.594 | 1.31 | 2.98 |
| OURS | **0.719** | **0.442** | **0.612** | **1.47** | **2.35** |

