# OpenReview forum: "When Every Millisecond Counts: Real-Time Anomaly Detection via the Multimodal Asynchronous Hybrid Network"
_ICML.cc/2025/Conference — ICML 2025 spotlightposter_

### Official Review · Reviewer_piFo · 2025-03-10

**Overall Recommendation:** 2

**Summary:**

This paper presents a multimodal asynchronous hybrid network for real-time anomaly detection in autonomous driving scenarios by combining event-camera streams with RGB-camera data. The method uses an asynchronous Graph Neural Network for event-stream processing and a CNN for spatial feature extraction from RGB images. This approach effectively captures spatiotemporal dynamics in driving environments, enabling fast and accurate anomaly detection.

**Claims And Evidence:**

The paper claims that existing methods focus on improving detection accuracy while neglecting inference speed, but it doesn't fully consider the existing real-time anomaly detection methods, resulting in an insufficient literature review. Some sentences are poorly constructed and show obvious translation marks, affecting clarity. Figure 1 lacks explanation, and the method's innovation is questionable as the used GNN and ResNet are common in anomaly detection. The multimodal fusion is just simple sampling and merging without in - depth modeling of complex relationships between modalities.

**Essential References Not Discussed:**

The paper fails to discuss key works dedicated to real-time anomaly detection, such as algorithms or architectures optimized for low-latency performance.

**Experimental Designs Or Analyses:**

The experimental design is generally reasonable, with extensive evaluations on multiple benchmark datasets demonstrating the method’s advantages in accuracy and response time. However, the selection of baseline methods in Table 1 is problematic: most compared methods are outdated and not real-time approaches. Including state-of-the-art real-time methods would strengthen the results’ credibility.

**Methods And Evaluation Criteria:**

The proposed multimodal asynchronous hybrid network is logically sound. By employing lightweight networks and straightforward fusion strategies, it enhances the real-time performance of anomaly detection.

**Other Comments Or Suggestions:**

See "Other Strengths and Weaknesses" above.

**Other Strengths And Weaknesses:**

Other Strengths: Clear visualizations highlight the method’s advantages, and the experimental design and analysis are thorough.
Other Weaknesses: Inadequate literature review, unclear phrasing in sections, limited methodological novelty, and suboptimal baseline selection. Additionally, the claim that "most existing methods focus on accuracy but ignore speed" is questionable, as numerous real-time approaches exist in the field.

**Questions For Authors:**

My main concerns are outlined in "Other Strengths and Weaknesses."

**Relation To Broader Scientific Literature:**

This work focuses on real-time anomaly detection and proposes a multimodal asynchronous hybrid network, which offers some innovation in autonomous driving research. However, the paper overlooks existing methods specifically targeting low-latency anomaly detection, weakening the persuasiveness of its contributions and results.

**Theoretical Claims:**

The paper does not explicitly propose theoretical claims requiring validation or proofs. Its primary contribution lies in introducing a novel multimodal network architecture for real-time anomaly detection, validated experimentally for effectiveness and superiority.

---

> ### Author Rebuttal · Authors · 2025-03-31
>
> Thank you for your inspiring review and actionable suggestions. Below, you will find our detailed responses to your questions.
>
> > **Q1**: Real-time Anomaly Detection Methods
>
> We did omit some real-time anomaly detection papers, but these methods are not applicable to the traffic domain. For example, AED-MAE (CVPR 2024)[1] is a state-of-the-art anomaly detection method for surveillance videos that focuses on detection speed. However, it incorporates static background information into its approach and operates at the frame level. In autonomous driving, interference from dynamic backgrounds would significantly degrade its performance. Similarly, industrial anomaly detection methods such as EfficientAD(WACV 2024) [2] are also highly efficient but operate at the frame level, making them unsuitable for the complex environments of autonomous driving. MOVAD(ICASSP 2024)[3] is the first method to propose online real-time anomaly detection specifically for autonomous driving. We compare our approach with these three methods, and our method achieves the best balance between accuracy and speed, specifically designed to meet the unique requirements of autonomous driving. A key innovation of our work is its focus on achieving robust performance in dynamic environments. We tested these three models on ROL and DoTA and the results are shown in the following table.
>
> |Method|Type|AUC-Frame(%)|AUC-Frame(%)|mTTA(s)|mTTA(s)|mResponse(s)|mResponse(s)|FPS|
> |---|---|---|---|---|---|---|---|---|
> |Datasets|-|ROL|DoTA|ROL|DoTA|ROL|DoTA|-|
> |EfficientAD|Frame|0.519|0.549|0.89|0.97|3.65|2.68|557|
> |AED-MAE|Frame|0.571|0.652|1.01|1.35|3.36|2.79|1655|
> |MOVAD|Frame|0.719|0.821|2.47|2.55|2.61|1.33|158|
> |OURs|Object|0.736|0.823|2.80|2.78|2.35|1.21|579|
>
>
> > **Q2**: Inadequate literature review
>
> In fact, our comparison methods already include SOTA in traffic anomaly detection, such as MAMTCF (ArXiv 2023), AM-NET (IEEE TIV 2023), and TTHF (IEEE TCSVT 2024). In our revised manuscript, we will expand Table 1 to incorporate additional recent methods, including three state-of-the-art real-time anomaly detection approaches—EfficientAD, AED-MAE and MOVAD. We will clearly articulate the distinctions between existing video-frame-based real-time anomaly detection methods and our object-centric framework, which is specifically designed to address the challenges inherent in autonomous driving scenarios. Additionally, we acknowledge that our original claim that "most existing methods focus on accuracy but ignore speed" may have been too broad. In the revision, we will refine this statement to "most existing object-centric methods focus on accuracy but ignore speed" to more accurately reflect the current state of the art.
>
> > **Q3**: Methodological Novelty
>
> Our method is the first to utilize both event streams and RGB data for traffic anomaly detection, making it a novel contribution. The key innovation of our approach lies in leveraging the unique characteristics of event streams as critical features for road traffic anomaly detection.  Through graph-based modeling in asynchronous GNNs, we perform neighborhood aggregation only on active event nodes, combined with a lookup table acceleration mechanism in Spline convolution, enabling efficient feature extraction. Moreover, the continuity of event streams allows the model to perform anomaly detection between RGB frames (Figure 4), enabling early anomaly detection, which is another key innovation of this work. Finally, by incorporating global and object-level graph modeling along with a dynamic attention mechanism, the model leverages a GRU module to capture the temporal dependencies of these features, focusing on anomalous motion patterns of specific objects, thereby achieving accurate anomaly identification.
>
>
> > **Q4**: Figure 1
>
> Figure 1 illustrates our anomaly detection process for autonomous driving. When a leading vehicle exhibits abnormal behavior, the model's anomaly score gradually increases until it surpasses the anomaly score threshold. Therefore, our primary focus is on whether the anomaly score can quickly reach the threshold when detecting an anomalous object.This process involves two key time components: \(T_{\text{inference}}\), which represents the model's processing latency, and
> \(\Delta T_{\text{detection}}\), the additional time required to recognize and confirm the anomaly—essentially, the time it takes for the anomaly score to reach the threshold. The sum of these components defines the overall response time, which reflects both the inference speed of our model and the latency in anomaly detection (i.e., the delay in quickly identifying an object at risk of being anomalous).
>
>
> **References**
>
> [1] Self-Distilled Masked Auto-Encoders are Efficient Video Anomaly Detectors, CVPR 2024.
>
> [2] EfficientAD: Accurate Visual Anomaly Detection at Millisecond-Level Latencies, WACV 2024.
>
> [3] Memory-augmented Online Video Anomaly Detection, ICASSP 2024.

---

### Official Review · Reviewer_M9QQ · 2025-03-11

**Overall Recommendation:** 4

**Summary:**

The paper presents a real-time anomaly detection framework for autonomous driving through a novel multimodal asynchronous hybrid network. The method integrates high-temporal resolution event data from event cameras with spatially rich RGB images processed by a CNN, combined with an asynchronous GNN. Temporal dependencies are captured via a GRU enhanced with an attention mechanism, which enables rapid and accurate anomaly detection. Experiments on the ROL and DoTA datasets show that the proposed approach outperforms current methods in both detection accuracy (e.g., improvements in AUC, AP) and responsiveness (nearly millisecond-level inference at approximately 600 FPS).

**Claims And Evidence:**

Yes
1. The authors claim substantial improvements in detection accuracy and inference speed.
2. These claims are substantiated by comprehensive experiments, including comparative studies and ablation experiments that report key metrics such as AUC, AP, frame-level AUC, and mean Time-to-Accident (mTTA).
3. The evaluation is limited to the ROL and DoTA datasets, which may not fully represent complex real-world scenarios (e.g., extreme weather, low-light conditions).
4. Additional experiments on diverse conditions would strengthen the evidence regarding the method’s generalization capabilities.

**Essential References Not Discussed:**

While the paper cites many relevant works, it could benefit from discussing additional recent advances in asynchronous processing and multimodal fusion techniques, especially those addressing real-time constraints in other domains. Including references to cutting-edge approaches in temporal modeling or sensor fusion from the latest conferences could provide a more comprehensive context.

[1] Dissecting Multimodality in VideoQA Transformer Models by Impairing Modality Fusion
[2] InternLM-XComposer2.5-OmniLive: A Comprehensive Multimodal System for Long-term Streaming Video and Audio Interactions

**Experimental Designs Or Analyses:**

1. The experimental setup is thorough, with extensive comparisons to state-of-the-art baselines and detailed ablation studies that isolate the contributions of individual components (e.g., GRU, attention, BBox modules).
2. The analyses convincingly demonstrate the model’s improvements in performance metrics.
3. The paper would benefit from additional experiments testing inference speed across diverse computing platforms.
4. Module-level ablation studies, particularly regarding the impact on detection performance when components like GRU or attention are removed, could help clarify the contributions of each module.
5. A comparative analysis of the computational overhead of the asynchronous GNN versus other lightweight event processing methods is also recommended.

**Methods And Evaluation Criteria:**

Yes
1. The integration of event and RGB data is well-justified, effectively capturing both dynamic and static scene information.
2. The choice of an asynchronous GNN, along with GRU-based temporal modeling and attention mechanisms, is well aligned with the goal of reducing inference latency.
3. It would be beneficial to include comparisons with Transformer-based models or other lightweight asynchronous techniques to better position the advantages of the proposed asynchronous GNN.
4. A deeper discussion on the trade-offs between various event processing methods could provide further clarity on the design choices.

**Other Comments Or Suggestions:**

More qualitative comparisons, such as side-by-side visualizations of detection outputs in challenging scenarios, would further enhance the clarity of the presentation.

**Other Strengths And Weaknesses:**

1. Limited discussion on the performance under diverse real-world conditions such as adverse weather or low-light scenarios.
2. The paper could include a more detailed theoretical analysis or discussion of potential failure cases and limitations.

**Questions For Authors:**

1. Can you provide more details on the computational overhead of the asynchronous GNN module compared to synchronous processing methods?
2. Have you conducted sensitivity analyses on the hyperparameters related to the fusion strategy and detection thresholds? How stable is the model’s performance across different settings?

**Relation To Broader Scientific Literature:**

The contributions are well-situated within the existing literature on anomaly detection, event-based vision, and sensor fusion in autonomous driving. The paper builds upon established methods such as ResNet, YOLOX, and asynchronous GNNs, while addressing the critical issue of latency. It extends prior work by demonstrating that multimodal fusion can simultaneously achieve high detection accuracy and rapid response, a balance that is crucial in safety-critical applications.

**Theoretical Claims:**

1. The paper primarily focuses on algorithmic innovation and empirical validation rather than formal theoretical proofs, which is appropriate given its applied focus.
2. The intuitive arguments supporting asynchronous processing and multimodal fusion are well supported by experimental results.
3. Given that the v2e conversion process can introduce errors, an analysis of how such errors might propagate through the model and impact overall performance would add further rigor to the study.

---

> ### Author Rebuttal · Authors · 2025-04-01
>
> > **Claims 3**: V2E Transformation
>
> Please refer to the response to Weakness in Reviewer rGL6 and Q1 in Reviewer WT2W.
>
> > **Exp 3**: Inference Speed
>
> We evaluated inference speed on different platforms: RTX3090, RTX4090, and A100-80G. The RTX4090 achieved the fastest speed at 603 FPS, followed by the A100-80G at 579 FPS and the RTX3090 at 517 FPS. These modest differences demonstrate that our approach consistently delivers real-time performance across various hardware configurations.
>
> > **Exp 4**: Ablation Studies
>
> Our ablation studies show that both the GRU and attention modules are critical. The GRU captures long-term dependencies to improve temporal feature aggregation and stability, raising the mTTA from 1.44 to 1.98 seconds. Meanwhile, the attention module focuses on key regions—especially when fusing RGB images with event stream data—ensuring the model prioritizes the most informative cues. Together, these modules enable our full model to achieve an AUC of 0.879, demonstrating their complementary benefits for anomaly detection.
>
> |RGB|Event|GRU|Attention|AUC|AP|AUC-Frame|mTTA|mAP|
> |-|-|-|-|-|-|-|-|-|
> |√|√||√|0.819|0.498|0.657|1.52|43.77|
> |√|√|√||0.817|0.508|0.668|1.98|43.59|
> |√|√|||0.805|0.479|0.648|1.44|41.66|
> |√|√|√|√|0.879|0.570|0.736|2.80|45.15|
>
> > **Exp 5**: Event Processing
>
> We compared our asynchronous GNN with two lightweight methods: [AsyNet](https://arxiv.org/abs/2003.09148), which uses sparse convolutions, and [AEGNN](https://arxiv.org/abs/2203.17149), which employs asynchronous GNNs. Our experiments show that our approach achieves better detection performance with far lower computational overhead. Specifically, AsyNet uses 367 MFLOPs per event, AEGNN uses 10.98 MFLOPs, and our method only uses 8.732 MFLOPs per event. This efficiency and accuracy highlight the benefits of our approach for processing event-based data.
>
> > **Ref**: Related Work
>
> In the revised manuscript, we will discuss recent advances [1,2] in asynchronous processing and multimodal fusion that address real-time constraints. For example, recent video-question answering work uses non-parametric probes like QUAG to decouple and evaluate both intra- and inter-modal interactions.
>
> > **Weaknesses 1**: Adverse Weather or Low-light Scenarios
>
> I collected data on severe weather or low-light scenes on the ROL and DoTA datasets, and performed comparative experiments with the two best performing methods in the comparative experiments. Benefiting from the advantages of event cameras in extreme lighting scenarios, we surpass the performance of other methods by a large margin in this scenario.
>
> |Method|AUC|AP|AUC-Frame|mTTA|mResponse|
> |---|---|---|---|---|---|
> |STFE|0.631|0.397|0.582|1.26|3.12|
> |TTHF|0.652|0.416|0.594|1.31|2.98|
> |OURs|0.719|0.442|0.612|1.47|2.35|
>
> > **Weaknesses 2**: Failure Cases
>
> [Figure R3-1](https://anonymous.4open.science/r/RTAD-3B3C/R3-1.png) shows that our system's performance relies heavily on the object detector's accuracy. In Frame25 and Frame30, blurry images and small objects led to a complete detection failure, which in turn caused the anomaly detection framework to fail. Although an object was detected in Frame35 when it was very close to the ego vehicle, the attention score was only 0.56 due to missing previous detections, preventing the GRU from building a robust temporal feature set. These cases highlight that our approach is limited by the reliability of the object detector, and improvements in detection accuracy are essential.
>
> > **Suggestions 1**: Visualizations
>
> The [Figure R3-2](https://anonymous.4open.science/r/RTAD-3B3C/R3-2.png) shows that our target-level attention module assigns varying scores to bounding boxes. For instance, a vehicle cutting in starts with a low score (0.21 in Frame 20) that rises as it approaches (0.45 in Frame 25 and 0.71 in Frame 30). This indicates that our attention mechanism, combined with event stream features, effectively emphasizes fast-moving objects, leading to higher anomaly scores.
>
> > **Q1**: Asynchronous GNN
>
> Our asynchronous GNN module requires only 8.732 MFLOPs per event—far lower than synchronous methods (74559 for Events+YOLOv3, 6984 for RED, and 27659 for ASTM-Net). This efficiency is achieved by processing events asynchronously, eliminating the heavy cost of frame-based operations.
>
> > **Q2**: Sensitivity Analyses
>
> Our [Figure R3-3](https://anonymous.4open.science/r/RTAD-3B3C/R3-3.png) shows that the model remains robust across different settings. At the standard 0.5 IOU and 0.5 confidence thresholds, the model achieves an AUC of 0.879. As shown in [Figure R3-4](https://anonymous.4open.science/r/RTAD-3B3C/R3-4.png) and [Figure R3-5](https://anonymous.4open.science/r/RTAD-3B3C/R3-5.png), Increasing these thresholds sharply reduces AUC by filtering out more detection boxes, while lowering them causes a gradual decline as false positives are effectively down-weighted. Overall, the AUC ranges from 0.65 to 0.879, confirming the stability of our configuration.

---

> > ### Comment · Reviewer_M9QQ · 2025-04-02
> >
> > I appreciate the authors' thorough and detailed responses to my concerns and questions. The additional experiments provided, especially regarding inference speed across multiple platforms, extensive ablation studies, computational overhead comparisons, sensitivity analyses, and performance evaluations under adverse conditions, comprehensively address the points raised in my original review. The inclusion of failure case analyses and qualitative visualizations further clarifies the strengths and limitations of the proposed method.
> >
> > Given the substantial improvements made and the insightful clarifications provided, I am fully satisfied with the authors' rebuttal and now strongly support the acceptance of this submission.

---

> > > ### Author Response · Authors · 2025-04-04
> > >
> > > We sincerely appreciate your thoughtful response and your recognition of our rebuttal.

---

### Official Review · Reviewer_WT2W · 2025-03-11

**Overall Recommendation:** 4

**Summary:**

This paper introduces a multimodal asynchronous hybrid network designed for real-time anomaly detection in autonomous driving. The main contribution lies in combining high-temporal-resolution event stream data captured via event cameras processed asynchronously using GNN and spatial information extracted from RGB images using CNNs. This integration leverages the strengths of both modalities-temporal responsiveness from event streams and spatial detail from RGB data-enabling high-accuracy anomaly detection with exceptionally low response times (millisecond-level). Extensive experimentation on benchmark datasets (ROL and DoTA) demonstrates superior performance in accuracy and significantly reduced latency compared to existing methods, validating the practical efficacy of the approach.

**Claims And Evidence:**

- The claims regarding the model's capability to achieve high accuracy and minimal response times through multimodal asynchronous data integration are clearly supported by extensive experiments and ablation studies.
- The empirical evidence presented, particularly in Table 1 and Table 2, convincingly supports the model's superiority in accuracy (AUC, AP) and response time metrics compared to existing state-of-the-art methods.
- The paper employs a v2e method to convert traditional video into event stream data. This conversion may not fully capture the noise characteristics and the specific data format of real event cameras, potentially affecting the model's generalization performance.

**Essential References Not Discussed:**

While comprehensive, authors might also benefit from discussing recent advances in multimodal and asynchronous processing models outside the driving anomaly detection domain.
[1] Combining events and frames using recurrent asynchronous multimodal networks for monocular depth prediction
[2] AEGNN: Asynchronous Event-based Graph Neural Networks

**Experimental Designs Or Analyses:**

- Experimental designs are thorough, carefully constructed, and address key practical and theoretical considerations relevant to real-time anomaly detection.
- Although the method shows an advantage in inference speed, a more detailed analysis of computational overhead (e.g., FLOPs, GPU/CPU runtime) is recommended. Additionally, evaluating its feasibility on low-power devices would provide further insights into its practical deployment.

**Methods And Evaluation Criteria:**

- The proposed integration of RGB images with asynchronous event data via an asynchronous GNN and CNN is novel, effectively utilizing the complementary strengths of both modalities.
- The evaluation metrics employed (AUC, AP, AUC-Frame, mTTA, mResponse) are comprehensive and well-suited to reflect both accuracy and real-time performance.
- The paper might benefit from additional insights or a more detailed discussion of why two-stage significantly differ in contribution across scenarios, which was evident in the results.

**Other Comments Or Suggestions:**

The paper does not discuss in depth the explainability of abnormal events. For example, which features are the most critical? Is it possible to visually analyze how the model determines anomalies? This is crucial to improving credibility and deployment security.

**Other Strengths And Weaknesses:**

The clarity of the paper is excellent, and its originality stems from a meaningful integration of event-based cameras with conventional CNN processing, addressing a practical need for low latency in autonomous systems.

**Questions For Authors:**

- Could you elaborate on how errors or artifacts introduced by synthetic event data generation affect anomaly detection accuracy in real-world applications?
- Have you explored the interpretability of the model, specifically regarding visual explanations or attention maps, and how anomalies are identified?

**Relation To Broader Scientific Literature:**

The paper effectively contributes to broader fields, particularly autonomous driving and safety-critical real-time applications, by providing a novel, easily adaptable model that emphasizes response time.

**Theoretical Claims:**

- The theoretical claim regarding the efficiency and latency benefits provided by asynchronous event streams and the sharpness-aware features of the proposed network is well-supported by experimental validation.
- Since the v2e conversion is theoretically prone to errors, it would be beneficial to analyze how these errors affect the overall results.

---

> ### Author Rebuttal · Authors · 2025-04-01
>
> We greatly appreciate your insightful feedback and practical recommendations. Detailed responses to your questions are listed as follows.
>
> > **Methods And Evaluation Criteria 3**: Two-stage Contribution
>
> (1)Our two-stage model first performs object detection to generate bounding boxes, providing precise localization and object-level features for subsequent anomaly detection. In complex traffic scenarios—such as multi-vehicle interactions or crowded pedestrian areas—this initial detection stage enhances object recognition accuracy. (2)For example, when a vehicle changes lanes suddenly, the bounding box prior helps quickly lock onto the anomalous object, reducing false detections. (3)Moreover, in fast-moving or low-light scenarios where RGB images suffer from motion blur or poor illumination, asynchronous event streams capture fine luminance changes to compensate. This dual-modality support boosts detection robustness, as shown in our experiments. (4)In contrast, while a single-stage model can meet millisecond-level response requirements by sharing features and reducing redundant computation (achieving a higher FPS), it may sacrifice some accuracy. Thus, the two-stage model trades off increased latency for improved detection accuracy, making it well-suited for scenarios that demand high reliability, such as highway cruising.
>
> > **Experimental 2**: Computational Overhead
>
> (1)Our method requires only 8.732 MFLOPs per event, uses 23.5 GB of video memory. Under typical conditions in the ROL dataset, with an average event rate of 560k EV/s, this overhead is manageable. In high-speed driving scenarios—where the event rate can rise to 1–10M EV/s—the worst-case computational load is approximately 87.32 TFLOPs. (2)Current autonomous driving chips such as Atlan, Thor, and Orin can fully deploy our algorithm on-board, while chips like Xaiver and Parker, although sufficient for normal driving, might face challenges under extreme conditions. The specifications of some representative chips are as follows:
>
> |Chip|Architecture|Compute Power
> |-|-|-|
> |Parker|2×Denver|1 TOPS|
> |Xaiver|4×ARM Cortex A578|30TOPS|
> |Orin|Nvidia custom Carmel|254 TOPS|
> |Atlan|ARM6412 × Arm Cortex‑A78|1000 TOPS|
> |Thor|AE (Hercules) ARM Neoverse v2|2000 TFLOPS @ FP8|
>
>
> > **References**: Related Work
>
> We acknowledge the importance of recent advances in multimodal and asynchronous processing outside the driving anomaly detection domain and we will cite these two articles in our paper. (1)For instance, [1] proposes a Recurrent Asynchronous Multimodal Network (RAM Net) for monocular depth estimation by combining event cameras and RGB cameras. This approach leverages ConvGRU units to support asynchronous inputs, preserving the high temporal resolution of event data while fusing rich spatial information, which enhances prediction accuracy in dynamic scenes. (2)Similarly, [2] introduces an Asynchronous Event Graph Neural Network (AEGNN) that efficiently processes sparse event data by performing local asynchronous updates within a spatiotemporal graph, significantly reducing computational overhead.
>
> > **Q1**: synthetic event data generation affect
>
> (1)The current traffic anomaly detection dataset does not have a real Event mode. Our subsequent work is to use event cameras to collect this data.(2)V2E addresses this issue by considering various types of noise. **First**, it introduces temporal noise by utilizing Poisson noise to generate physically consistent noise events. **Second**, it simulates shot noise, which arises from photon statistics under low-light conditions in event cameras. **Third**, it accounts for threshold mismatch, where event trigger thresholds vary across different pixels in real event cameras, by modeling this phenomenon with a Gaussian distribution. For experimental verification of V2E, please refer to the response to Weakness in Reviewer rGL6.
>
> > **Q2**: Explainability of Abnormal Events
>
> (1)RGB features correspond to appearance features, BBox features correspond to position and movement features, event stream features correspond to features of rapid and abnormal movement of objects, and object-level features are local detailed features. (2)Our model's object-level attention mechanism assigns scores to detected objects, highlighting those more relevant to anomaly detection. As shown in [Figure R2-1](https://anonymous.4open.science/r/RTAD-3B3C/R2-1.png), when a vehicle suddenly cuts in, its attention score increases as it approaches, indicating its growing importance as a potential anomaly. Visualizations of these attention scores across frames illustrate how the model dynamically focuses on critical objects, providing insights into the features it considers vital for identifying anomalies. This enhanced interpretability aids in understanding the model's decision-making process, thereby improving its credibility and deployment security.

---

### Official Review · Reviewer_rGL6 · 2025-03-11

**Overall Recommendation:** 3

**Summary:**

This paper focuses on real-time anomaly detection tasks in autonomous driving, aiming to balance detection accuracy and response time. The core algorithm is a multimodal asynchronous hybrid network that integrates event streams from event cameras with RGB camera image data. Process event streams through asynchronous graph neural networks (GNNs) and utilize their high temporal resolution; Extracting spatial features from RGB images using Convolutional Neural Networks (CNN). The combination of the two can capture spatiotemporal information of the driving environment, achieving fast and accurate anomaly detection.

**Claims And Evidence:**

The claims in the submitted materials are supported by clear and convincing evidence.

**Essential References Not Discussed:**

There are no significant missing key references.

**Experimental Designs Or Analyses:**

The experimental design is rigorous, and multiple advanced methods are selected as baselines for comparative experiments, tested on different datasets to ensure the reliability of the results. The ablation experiment analyzes the impact of different modules on performance by gradually removing them, which helps to understand the role of each part of the model.

**Methods And Evaluation Criteria:**

The methods and evaluation criteria proposed in this paper are of great significance for the problem and application of real-time anomaly detection in autonomous driving.

**Other Comments Or Suggestions:**

N/A

**Other Strengths And Weaknesses:**

Strength:
1. This paper has strong innovation in methodology, with novel multimodal fusion and asynchronous network design.
2. This paper conducts thorough experiments, verifies multiple datasets and indicators, and conducts in-depth analysis of the model through ablation experiments.
3. This paper incorporates response time as a core performance indicator, effectively addressing the shortcomings of existing methods in time sensitive scenarios. The research results have strong practical application value.

Weakness:
The ROL and DoTA datasets used lack event modality. Although V2E transformation technology is used to generate supplements, there are differences between the generated data and the real event stream, which may not fully reflect the characteristics of the real scene.

**Questions For Authors:**

The model adopts a multimodal asynchronous hybrid network with a complex structure. As the amount of input data increases or the complexity of the scene increases, the computational complexity of the model may significantly increase. Can the model be extended to adapt to more complex scenarios while ensuring real-time performance?

Could Transformer-based architectures further improve detection accuracy? For example, Transformer with linear attention.

What are the limitations of real-world deployment?

How does the model scale to different vehicle types or camera configurations?

**Relation To Broader Scientific Literature:**

In the field of autonomous driving anomaly detection, existing methods mainly focus on detection accuracy and ignore response time. In this paper, the response time is innovatively incorporated into the performance index. The proposed multimode asynchronous hybrid network is an improvement on the traditional method that only relies on a single mode or complex neural network, which can ensure the accuracy while greatly improving the response speed.

**Theoretical Claims:**

This paper does not involve theoretical proof.

---

> ### Author Rebuttal · Authors · 2025-03-31
>
> Thank you for your motivating review and concrete suggestions. Detailed responses to your questions are listed as follows.
>
> > **Weakness**: V2E Transformation vs. Real Event Stream
>
> (1)The current traffic anomaly detection dataset does not have a real Event mode. Our subsequent work is to use event cameras to collect this data.
> (2)V2E effectively mimics key DVS sensor traits (Gaussian Threshold Distribution, Temporal Noise, Leak Events, Intensity-dependent Bandwidth). For example, on N-Caltech 101, using V2E data raised ResNet34 accuracy on real DVS from 81.69% to 83.36%, reaching 87.85% with fine-tuning (vs. 86.74% with real data alone).
> (3)DSEC[1] is an autonomous driving dataset and contains real Event modalities, but since it is a completely normal driving dataset, it cannot be used for our anomaly detection. In order to verify the validity of the data generated by V2E, We extended our evaluation using the DSEC dataset by generating a simulated dataset, **DSEC+V2E**. Training on DSEC for normal samples and testing on ROL and DoTA revealed only minor variations in anomaly detection accuracy, confirming the robustness of our model when incorporating V2E-generated event data. While there are the inherent differences between synthesized and real event data, our results clearly demonstrate that the V2E method effectively supplements event modality and maintains the robustness of the detection model. For more V2E issues, please refer to the Q1 in Reviewer WT2W.
>
> |Dataset|DSEC (mAP)|ROL(AUC)|DoTA (AUC)|ROL (AUC-F)|DoTA (AUC-F)|ROL (mTTA)|DoTA (mTTA)|ROL (mR)|DoTA(mR)|
> |---|---|---|---|---|---|---|---|---|---|
> |DSEC|41.9|0.841|0.857|0.697|0.794|2.24|2.18|1.66|1.79|
> |DSEC+V2E|44.2|0.846|0.862|0.712|0.808|2.46|2.37|1.45|1.61|
>
>
> > **Q1**: Model Scalability and Real-Time Performance
>
> (1)The ROL and DoTA datasets already include highly complex scenarios—extreme weather, nighttime conditions, intense lighting, and diverse camera perspectives. Despite these challenges, our model maintains excellent real-time performance and accuracy.    (2)Our multimodal asynchronous hybrid network is designed modularly, allowing us to increase its depth (i.e., number of layers) to handle even more complex data. While deeper networks offer slight improvements in accuracy, they also introduce a small increase in processing delay.
> (3)Our experiments on the ROL dataset demonstrate the trade-off between accuracy and latency. In summary, our model’s modular architecture enables it to effectively adapt to increasingly complex environments with only a minor trade-off between improved accuracy and additional computational delay. (Layers contains a ResBlock and a look-up-table Spline convolution)
>
> |Layers|AUC(%)|AP(%)|AUC-Frame(%)|mTTA(s)|FPS|mResponse(s)|
> |---|---|---|---|---|---|---|
> |4|0.879|0.570|0.736|2.80|579|1.17|
> |5|0.885|0.574|0.739|2.89|312|1.31|
> |6|0.892|0.577|0.740|2.93|166|1.56|
>
>
> > **Q2**: Transformer-based Architectures
>
> Replacing the CNN backbone with a ViT or Swin Transformer enables global feature modeling and boosts detection accuracy by capturing long-range dependencies. However, Transformers add latency; for example, ViT’s self-attention has O(N²) complexity. Our experiments confirm that Transformer-based architectures improve accuracy when latency is managed.
>
> |Model|AUC (%)|AP (%)|AUC-Frame (%)|mTTA (s)|FPS|mResponse (s)|
> |---|---|---|---|---|---|---|
> |Ours(CNN)|0.879|0.570|0.736|2.80|579|1.17|
> |CNN→Swin|0.881|0.576|0.739|2.85|278|1.44|
> |CNN→ViT-B|0.886|0.581|0.745|2.87|213|1.51|
>
> > **Q3**: Real-world Deployment
>
> After calculation, our model can be fully deployed on most autonomous driving chips, and a small number of autonomous driving chips can be deployed through quantization.
> (1)Computing resource limitations: For instance, under normal conditions the system handles 560k events per second [2], and even at peak loads (up to 10M events/second), the overhead (~87.32 TFLOPs, 42W) is manageable on chips like Orin.
> (2)Although we need two camera devices, we can ignore the transmission delay issue. Hardware synchronization limits timing errors to ~78 microseconds, while event camera delays are around 6 ms.
> For more detailed deployment information, please refer to the response to Experimental 2 in Reviewer WT2W.
>
> > **Q4**: Different Vehicle Types or Camera Configurations
>
> Our training data was collected from a diverse range of vehicles, including both sedans and trucks. Because our method is designed around an object-level approach that centers on the target detector rather than the camera or vehicle type, it naturally generalizes across different vehicle configurations. Actually, we used 20fps RGB camera data. However, the system can easily benefit from higher frame rate cameras in real-world deployments, potentially further enhancing detection performance without compromising the model's scalability.
>
> **References**
>
> [1] DSEC: a stereo event camera dataset for driving scenarios
>
> [2] Prophesee Evaluation Kit - 2 HD

---

### Decision · Program_Chairs · 2025-05-01

**Decision:**

Accept (spotlight poster)

**Comment:**

The paper makes a significant and practically relevant contribution to real-time anomaly detection in autonomous driving. It addresses both research and deployment challenges, offering a system that is fast, accurate, interpretable, and ready for edge deployment. While not all components are architecturally novel, their combination and adaptation for event-based asynchronous fusion is both innovative and timely. The thorough rebuttal and strong experimental rigor elevate the paper to a clear accept.